# Structure insights into selective coupling of G protein subtypes by a class B G protein-coupled receptor

Li-Hua Zhao [1,2,9] ✉, Jingyu Lin [3,9], Su-Yu Ji[4,9], X. Edward Zhou[5,9], Chunyou Mao [4], Dan-Dan Shen [4], Xinheng He[1,2], Peng Xiao [3], Jinpeng Sun [3], Karsten Melcher[5], Yan Zhang [4,6,7,8] ✉, Xiao Yu [3] ✉ & H. Eric Xu [1,2] ✉

The ability to couple with multiple G protein subtypes, such as $G_s$, $G_{i/o}$, or $G_{q/11}$, by a given G protein-coupled receptor (GPCR) is critical for many physiological processes. Over the past few years, the cryo-EM structures for all 15 members of the medically important class B GPCRs, all in complex with $G_s$ protein, have been determined. However, no structure of class B GPCRs with $G_{q/11}$ has been solved to date, limiting our understanding of the precise mechanisms of G protein coupling selectivity. Here we report the structures of corticotropin releasing factor receptor 2 (CRF2R) bound to Urocortin 1 (UCN1), coupled with different classes of heterotrimeric G proteins, $G_{11}$ and $G_o$. We compare these structures with the structure of CRF2R in complex with $G_s$ to uncover the structural differences that determine the selective coupling of G protein subtypes by CRF2R. These results provide important insights into the structural basis for the ability of CRF2R to couple with multiple G protein subtypes.

G protein-coupled receptors (GPCRs) comprise a large and diverse family of cell-surface receptors, with over 800 members encoded in the human genome[1]. In general, GPCRs are activated by ligands and then the active GPCRs regulate diverse physiological processes through activation of heterotrimeric G proteins and other intracellular effectors[2]. There are four major subtypes of heterotrimeric G proteins (Gαβγ), typified by their Gα subunit: Gα$_s$, Gα$_{i/o}$, Gα$_{q/11}$, and Gα$_{12/13}$[3–9]. Many GPCRs can couple with more than one subtype of G protein, each with a distinct coupling profile that evokes a unique cellular response, which is defined as GPCR biased activation[10–13]. Determining the basis

for specific GPCR coupling profiles is critical to understanding their biology and pharmacology.

Corticotropin releasing factor (CRF) and three urocortin peptides (UCN1, UCN2, UCN3) are crucial stress hormones that can differentially bind to and activate CRF receptors type 1 (CRF1R) and type 2 (CRF2R), which are members of the class B GPCRs. Both CRF1R and CRF2R are thought to mediate diverse signaling pathways through their interactions with several heterotrimeric (αβγ) G protein subtypes, including different Gα subunits, such as Gα$_s$, Gα$_i$, Gα$_o$, Gα$_q$, and Gα$_{11}$[8,9,14–17]. CRF1R and CRF2R primarily activate cAMP-PKA pathways via G$_s$

[1]The CAS Key Laboratory of Receptor Research, Shanghai Institute of Materia Medica, Chinese Academy of Sciences, Shanghai 201203, China. [2]University of Chinese Academy of Sciences, Beijing 100049, China. [3]Department of Physiology, School of Basic Medical Sciences, Shandong University, Jinan 250012, China. [4]Department of Biophysics and Pathology of Sir Run Run Shaw Hospital, Zhejiang University School of Medicine, Hangzhou 310058, China. [5]Department of Structural Biology, Van Andel Research Institute, Grand Rapids, MI 49503, USA. [6]Liangzhu Laboratory, Zhejiang University Medical Center, Hangzhou 311121, China. [7]MOE Frontier Science Center for Brain Research and Brain-Machine Integration, Zhejiang University School of Medicine, Hangzhou 310058, China. [8]Zhejiang Provincial Key Laboratory of Immunity and Inflammatory diseases, Hangzhou 310058, China. [9]These authors contributed equally: Li-Hua Zhao, Jingyu Lin, Su-Yu Ji, X. Edward Zhou. ✉e-mail: zhaolihuawendy@simm.ac.cn; zhang_yan@zju.edu.cn; yuxiao@sdu.edu.cn; eric.xu@simm.ac.cn

coupling, which mediate stress responses and have been implicated in the pathophysiology of various diseases[14,18,19].

UCN1 is a high affinity ligand for both CRF1R and CRF2R[14,20]. We have previously reported cryo-EM structures of UCN1-bound CRF1R and CRF2R in complex with a heterotrimeric $G_s$ protein[21]. In addition, both CRF1R and CRF2R have been shown to couple with G protein subtypes of $G_{q/11}$ and $G_{i/o}$[8,14,18]. Different G protein subtypes downstream of CRF1R and CRF2R were known to couple with distinct functions. For instance, the $G_s$-PKA signaling was reported to mediate the CRF2R function in promoting lipolysis metabolism[22]. In contrast, during pregnancy and labour, CRF2R couples with $G_q$ to promote myometrial contractility and quiescence via activation of ERK and PKC pathways[23,24]. Moreover, coupling of CRF1R to $G_i$ enabled Src activation and signaling of downstream Akt and ERK, which may participate in anxiety, depression and stress responsiveness[25-28]. However, the molecular mechanism underlying the ability of CRF2R to couple with multiple G protein subtypes remains unclear due to the lack of CRF2R structures in complex with G protein subtypes of $G_{q/11}$ and $G_{i/o}$.

In this paper, we overcome technical challenges to assemble stable complexes of UCN1-bound CRF2R with G protein subtypes of $G_{11}$ and $G_o$, and determine their cryo-EM structures. Our results provide detailed structural insights into the ability of class B GPCRs to interact with multiple G protein subtypes[29].

## Results

### Cryo-EM structure determination of UCN1-CRF2R-$G_{11}$ and UCN1-CRF2R-$G_o$ complexes

To prepare high quality human UCN1-CRF2R-$G_{11}$ and UCN1-CRF2R-$G_o$ complexes, we developed the NanoBiT tethering strategy to stabilize the complexes[30-32] (Supplementary Fig. 1a, b). We used dominant negative $DNG\alpha_{11}$ and $DNG\alpha_o$, which are modified forms of $G\alpha_{11}$ and $G\alpha_o$ that have two amino acid replacements equivalent to those in a published dominant-negative bovine $G\alpha_s$ (DNG$\alpha$s) construct[33]. In addition, $DNG\alpha_{11}$ (residues 1-24) is replaced with $G\alpha_{i1}$ (residues 1-18) and $DNG\alpha_o$ (residues 1-29) is replaced with $G\alpha_{i1}$ (1-29) to possess the ability to bind scFv16[34]. Large-scale purification was performed to obtain the UCN1-CRF2R-$G_{11}$ and UCN1-CRF2R-$G_o$ complexes for cryo-EM studies (Supplementary Fig. 1a, b).

The images of UCN1-CRF2R-$G_{11}$ and UCN1-CRF2R-$G_o$ complexes were collected by a Titan Krios with a Gatan K3 detector and a Titan Krios with a Gatan K2 detector, respectively (Supplementary Figs. 2a and 3a). 2D classification showed clear secondary structure features and random distribution of the particles, which enabled a high-resolution reconstruction of the cryo-EM density maps (Supplementary Figs. 2b and 3b). The structures of the UCN1-CRF2R-$G_{11}$ and UCN1-CRF2R-$G_o$ complexes were determined from a total of 3,402,020 and 1,840,659 initial particles to an overall resolution of 3.7 Å and 2.8 Å, respectively (Fig. 1, Supplementary Figs. 2c, d, 3c, d and 4a, b). Both

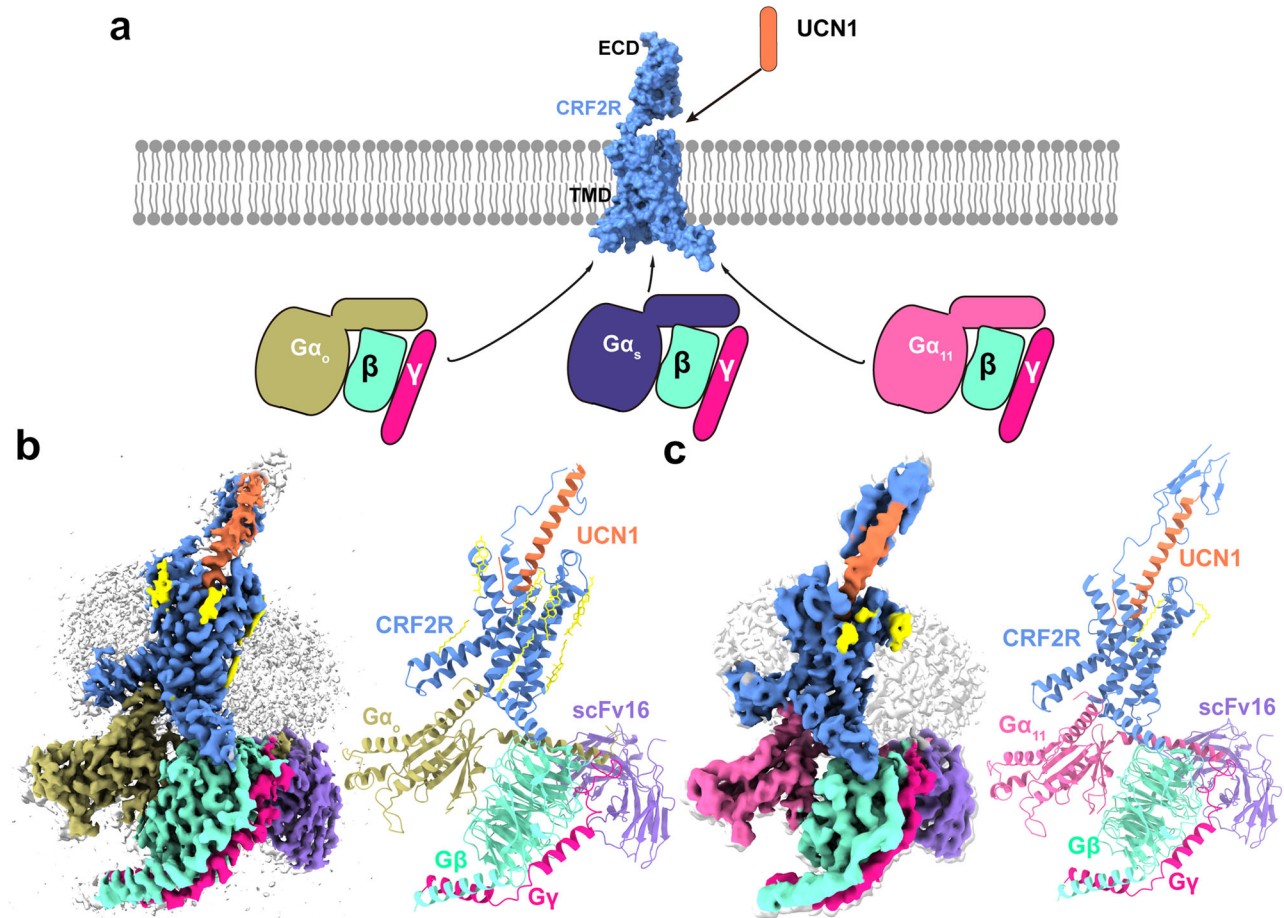

**Fig. 1 | The overall cryo-EM structures of UCN1-CRF2R-G protein complexes.**
**a** Cartoon representation of signaling selectivity of CRF2R. **b** Left, cut-through view of the cryo-EM density map of the UCN1-CRF2R-$G_o$ complex and the disc-shaped micelle. The unsharpened cryo-EM density map at the 0.016 threshold shown as light gray surface indicates a micelle diameter of 11 nm. The colored cryo-EM density map is shown at the 0.022 threshold. Right, cartoon representation of the UCN1-CRF2R-$G_o$ complex is shown with annular lipids in yellow stick representation. Cornflower blue, CRF2R; coral, UCN1; dark khaki, $G_o$; aquamarine, G$\beta$; deep pink, G$\gamma$; medium purple, scFv16. **c** Left, cut-through view of the cryo-EM density map that represents the UCN1-CRF2R-$G_{11}$ complex and the disc-shaped micelle. The unsharpened cryo-EM density map at the 0.013 threshold shown as light gray surface indicates a micelle diameter of 11 nm. The colored cryo-EM density map is shown at the 0.016 threshold. Right, cartoon representation of the UCN1-CRF2R-$G_{11}$ complex shown with annular lipids in yellow stick representation.

structures were carefully examined, and refined with real space refinement in Phenix[35]. The model of the CRF2R-G$_{11}$ complex were further refined with Rosetta refinement techniques against the cryo-EM map (Supplementary Fig. 4c–f)[36]. The Rosetta refinement improved the atomic details in the CRF2R-G$_{11}$ complex structure that was built in a 3.7 Å cryo-EM map, with chemically optimized side chain rotamers and the global protein geometry (Supplementary Data 1). We truncated many side chains in the Rosetta-refined model of the CRF2R-G$_{11}$ complex to obtain the final model for Protein Data Bank deposition, to reflect the quality of the electron density map. However, we used the side chain rotamers from the Rosetta-refined model for the purpose of discussion herein, which are indicated in the relevant figure legends.

## Comparison of overall structures

The overall structures of the UCN1-CRF2R-G$_{11}$ and UCN1-CRF2R-G$_o$ complexes are similar to the previous structure of the UCN1-CRF2R-G$_s$ complex (PDB: 6PB1)[21], with root mean square deviation (RMSD) values of 1.01 Å and 0.64 Å for the Cα atoms of the receptor. We observed structural differences in the second extracellular loop (ECL2), the second intracellular loop (ICL2), helix 8 (H8) of the receptor and the Gβ subunit (Fig. 2), the αN and α5 helices of Gα in these complexes (Fig. 3). Although UCN1 binds at a very similar site (Fig. 2a–c and e), the ECL2 is closer to the N terminus of UCN1 in the UCN1-CRF2R-G$_{11}$ structure. The main chain carbonyl group of P3$^{UCN1}$ forms an H-bond with the side-chain of K258$^{ECL2}$, which was modeled with Rosetta in the UCN1-CRF2R-G$_{11}$ structure and P3$^{UCN1}$ also forms an H-bond with the side-chain of K258$^{ECL2}$ in UCN1-CRF2R-G$_o$ structure (Fig. 2c and Supplementary Fig. 4c). Consistently, alanine mutation of K258$^{ECL2}$ diminished the receptor coupling to G$_{11}$ and G$_o$ compared with wild-type (WT) CRF2R, while this mutation does not affect G$_s$ coupling of the receptor (Supplementary Fig. 5 b, e, h and Table 2). The main changes in the receptor are the different conformations of ICL2, whose conformation is critical for mediating G protein recognition and specificity (Fig. 2d), as discussed later (Fig. 4).

Compared with the UCN1-CRF2R-G$_s$ structure, H8 of CRF2R moved 2.5 Å and 2.1 Å in the opposite direction between the UCN1-CRF2R-G$_{11}$ and UCN1-CRF2R-G$_o$ structures (Fig. 2f–i). The interactions between the C terminus of CRF2R and the Gβ subunit were also observed in these structures. The side chain of D379$^{8.63b}$, R376$^{8.60b}$ and K372$^{8.56b}$ in H8 forms a hydrogen bond with R304, H311 and D312 in the Gβ subunit from the CRF2R-G$_o$ structure, respectively. While the side chain of R376$^{8.60b}$ and the main chain carbonyl of D379$^{8.63b}$ in H8 forms a hydrogen bond with D312 and R42 in the Gβ subunit from the CRF2R-G$_s$ structure, respectively. In the CRF2R-G$_{11}$ structure, possibly due to poor map quality, only one hydrogen bond was observed between the side chain of R376$^{8.60b}$ in H8 and D312 in the Gβ subunit (Fig. 2f–i, Supplementary Fig. 4d).

Besides our CRF2R-G$_{11}$ complex structure, G$_{11}$-coupled muscarinic acetylcholine receptor 1 (M1R) and human cytomegalovirus (HCMV) encodes GPCR US28 are the only two available G$_{11}$-bound GPCR structures (PDB: 6OIJ and 7RKF)[37,38]. Only US28-G$_{11}$ is in GDP-bound form, while both CRF2R-G$_{11}$ and M1R-G$_{11}$ complexes are nucleotide-free. The nucleotide-free G$_{11}$ in M1R-G$_{11}$ and CRF2R-G$_{11}$ complexes share several common structural features, but show clear differences in conformational details between each other. To analyze how G$_{11}$ couples to largely different class A M1R and class B CRF2R, we compared our CRF2R-G$_{11}$ structure with the M1R-G$_{11}$ complex structure. We observed that TM6 and TM7 of the two receptors adopt largely different conformations in their G$_{11}$ complexes. In contrast, ICL2 of both receptors form a similar helix when binding to G$_{11}$ (Fig. 5a–c), which extensively interact with the Gα-αN helix and Gα-α5 helix of the G protein. In the CRF2R-G$_{11}$ structure, E220$^{ICL2}$ in ICL2 interacts weakly with R37 at the C-terminal end of the αN helix of G$_{11}$, while R134$^{ICL2}$ in ICL2 of M1R forms strong hydrogen bond with R37 of G$_{11}$. In addition, both Y217$^{ICL2}$ of the CRF2R and the related L131$^{ICL2}$ of M1R interact with

the β2-β3 loop and Gα-α5 helix of G$_{11}$ (Fig. 5b, Supplementary Fig. 4f). We also found differences in the C termini of the receptors and the Gβ subunit in the CRF2R-G$_{11}$ and M1R-G$_{11}$ complexes. In the M1R-G$_{11}$ complex, the C terminus after H8 is extended into a groove formed by the Ras domain of G$_{11}$ and the Gβ (Fig. 5d, e)[37]. CRF2R had a polybasic cluster in H8 that shows different mode of interaction with Gβ (Fig. 5d, e). It seems that M1R has a polybasic C-terminal cluster to engage G$_{q/11}$ subtype of G proteins more efficiently than class B GPCRs[37]. In addition, there are some conformational changes of the Gα-αN and Gα-α5 helices relative to the receptor (Figs. 2d, 3 and 4), which may be important for receptor activation and transducer coupling in different Gα-bound complexes.

## Comparison of the G protein-binding pockets

In the UCN1-CRF2R–G protein structures, an outward shift of TM6 at the intracellular side of the receptor forms a sharp kink, generating a common binding pocket for G protein coupling, where the C terminus of the Gα-α5 helix binds to the receptor (Fig. 3a, b). Although CRF2R shares a common binding pocket for coupling to different Gα subunits, the ability of coupling is different. The C-terminus of the α5 helix is widely considered to be the most important structural determinants of G protein coupling selectivity[39,40]. Comparison of these three structures shows a clear difference in the orientation of the α5 helix of G$_{11}$, G$_o$ and G$_s$ relative to CRF2R. The α5 helix of G$_o$ is rotated -8.2° away from receptor ICL2 and the α5 helix of G$_{11}$ is rotated -2.8° towards receptor ICL2 compared to this of G$_s$ (Fig. 4a). The sequences of the C-termini of the Gα-α5 are different among these G protein subtypes (Fig. 3c). The third and fourth to last residues are Y391 and E392 in G$_s$, C351 and G352 in G$_{i/o}$, and Y356 and N357 in G$_{q/11}$. These residues are less conserved among G protein subtypes and are located at the interface with TM5-6, the C-terminus of TM7 and the N-terminus of H8 of the receptor. The measured interaction interface formed between CRF2R and the α5 C terminus (residues G.H5.16 to G.H5.26) is larger for G$_s$ (792.7 Å$^2$) than for G$_{11}$ (524.7 Å$^2$) and G$_o$ (447.7 Å$^2$) (Fig. 3d–f). The bulkier side chains of Y391 and E392 in G$_s$ can form the largest interaction interface between CRF2R and the G$_s$-α5 C terminus (Fig. 3g–i). Therefore, class B GPCRs perform their physiological actions primarily by coupling to G$_s$. It can be explained by the fact that the G protein–binding pocket of class B GPCRs prefers to accommodate primarily the bulkier C-terminus of the α5 in G$_s$, but can still couple to the less bulkier C-terminus of the G$_{q/11}$-α5 and G$_{i/o}$-α5[41], consistent with the importance of the Gα C-terminus for G protein selectivity determinants[39].

To accommodate the Gα-α5 helix, the cytoplasmic end of TM6 has a sharp kink as observed in all class B GPCR–G complex structures[21,30–33,41–48], including the CRF2R-G$_{11}$ and CRF2R-G$_o$ complexes (Fig. 3b), and the GCGR-G$_i$ structures (PDB: 6LML)[41] (Fig. 6g). The interface residues from the receptor cytoplasmic cavity that contact the C terminus of the α5 helix are highly conserved among class B GPCRs, suggesting a common mechanism of G protein coupling by class B GPCRs. Comparison of the CRF2R-G$_{11}$ and CRF2R-G$_o$ structures with class A GPCR structures, including the M1R-G$_{11}$ structure (PDB: 6OIJ)[37], the M2R-G$_o$ structure (PDB: 6OIK)[37] and the 5HT$_{1B}$-mG$_o$ structure (PDB: 6G79)[49], reveal different positioning of TM6 among these complexes (Figs. 5c and 6h–i). The different conformations of TM6 in these receptors allow different anchoring of the α5 helix of Gα into their distinct cytoplasmic pockets[41] (Figs. 5 and 6a, b).

## Conformational differences in ICL2 determine G protein recognition and specificity

For most receptors, coupling selectivity is mainly determined by the Gα-α5 helix and the Gα subunit core[39,40]. In addition to the α5 helix, we also observe differences in the position of the Gα-αN and β1 strand of Gα, both elements interact with ICL2 of the receptor. Compared with the G$_s$-bound structure, the αN helix and β1 strand of Gα shifted away

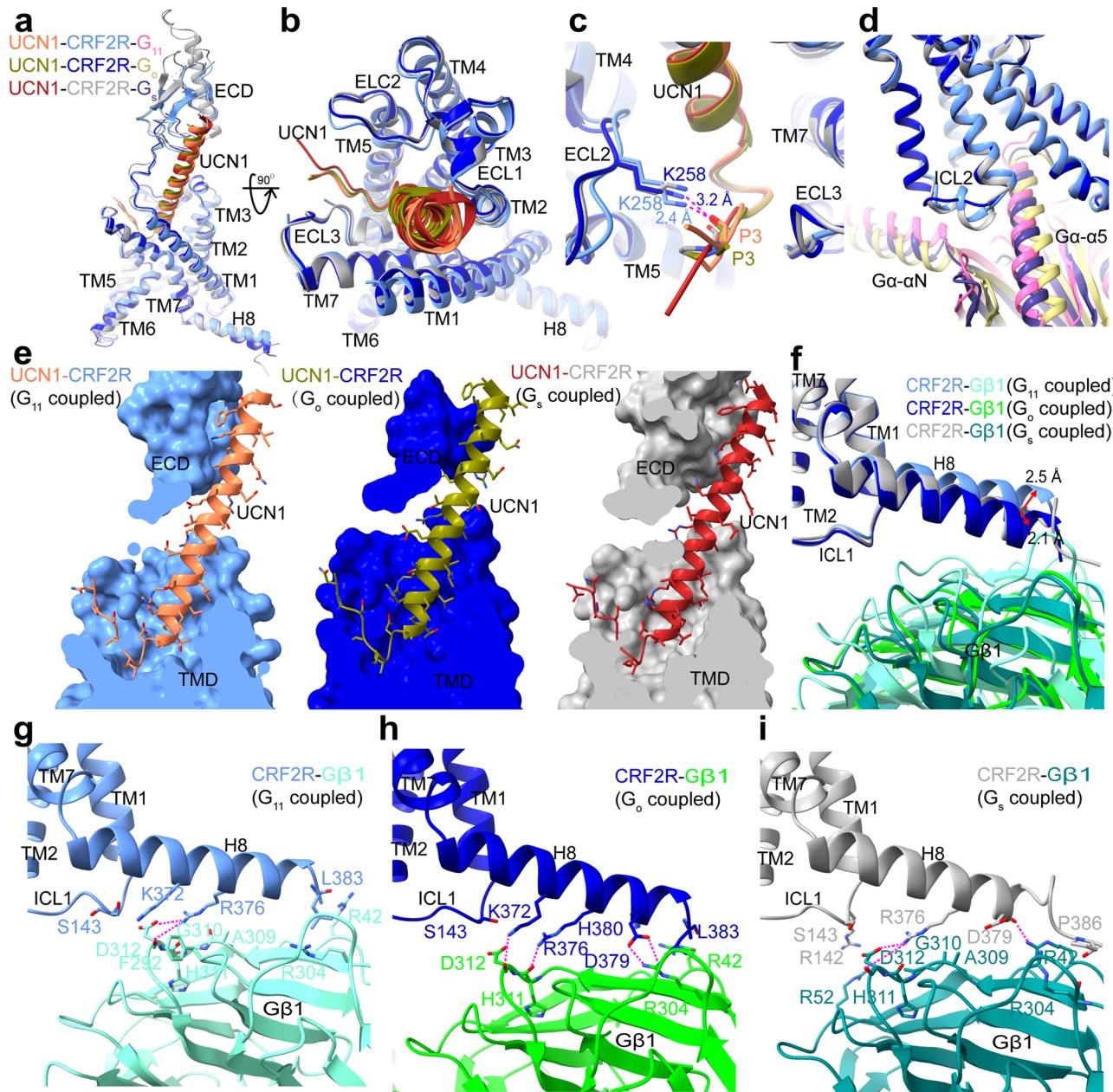

**Fig. 2 | Universal and unique aspects of different G protein coupling by CRF2R.**
**a** Comparison of UCN1 and CRF2R in the G protein-bound structures in the side view. **b** Comparison of UCN1 and the TMD conformation in the G protein-bound structures in an extracellular view. **c** Comparison of the N terminus of UCN1 and the position of the ECL2 of CRF2R in the G protein-bound structures. **d** Comparison of the distinct conformational changes of ICL2 and the differences in the orientation of Gα in these three UCN1-CRF2R-G protein structures. **e** The binding pocket of UCN1(coral) in CRF2R (cornflower blue)-$G_{11}$ (hot pink), UCN1(olive) in CRF2R (medium blue)-$G_o$ (dark khaki), and UCN1 (fire brick) in CRF2R (dark gray)-$G_s$ (dark

slate Blue). Many UCN1 side chains in the UCN1-CRF2R-$G_{11}$ structure were truncated, whose rotamers shown here were based on the Rosetta-refined model.
**f** Structural comparison of CRF2R H8 and Gβ1. **g** CRF2R H8 (cornflower blue)-Gβ1 (aquamarine) interface in the UCN1-CRF2R-$G_{11}$ complex. The side chains of K372, and D379 of the receptor, and R42 and D312 of Gβ1were truncated in the structure, whose rotamers shown in this panel was from the Rosetta-refined model. **h** CRF2R H8 (medium blue)-Gβ1 (lime green) interface in the UCN1-CRF2R-$G_o$ complex. **i** CRF2R H8 (dark gray)-Gβ1 (teal) interface in the UCN1-CRF2R-$G_s$ complex (PDB: 6PB1). The polar contacts are shown as purple dashed lines.

from the receptor ICL2 in the $G_o$-bound structure and shifted towards the receptor ICL2 in the $G_{11}$-bound structure (Fig. 4a). This movement are associated with the different conformations of the Gα-ICL2 interfaces of the receptor in the complexes with different G protein subtypes. The receptor ICL2 forms the most extensive interactions with $G_s$ and the least interaction with $G_o$, in receptor-G protein complexes (Fig. 4c–e). The distinct conformational changes of ICL2 in these three UCN1-CRF2R-G protein structures may play key roles in G protein recognition and specificity.

Comparison of the three UCN1-CRF2R-G protein structures shows clearly different conformations of ICL2 that are likely induced by the different Gα-ICL2 interfaces. ICL2 in the CRF2R-$G_{11}$ and CRF2R-$G_s$ structures adopts a short helix that forms extensive interactions with $G_{11}$ and $G_s$ (Fig. 4a, c, d). The ICL2 of the receptor in the $G_o$ complex, however, adopts a loop and forms a smaller interface with $G_o$. While ICL2 of the receptor in $G_o$ complex is four residues shorter, the receptor TM4 is one helical turn longer than that in the complexes of $G_s$ and $G_{11}$. The extended TM4 in the CRF2R-$G_o$ structure provides

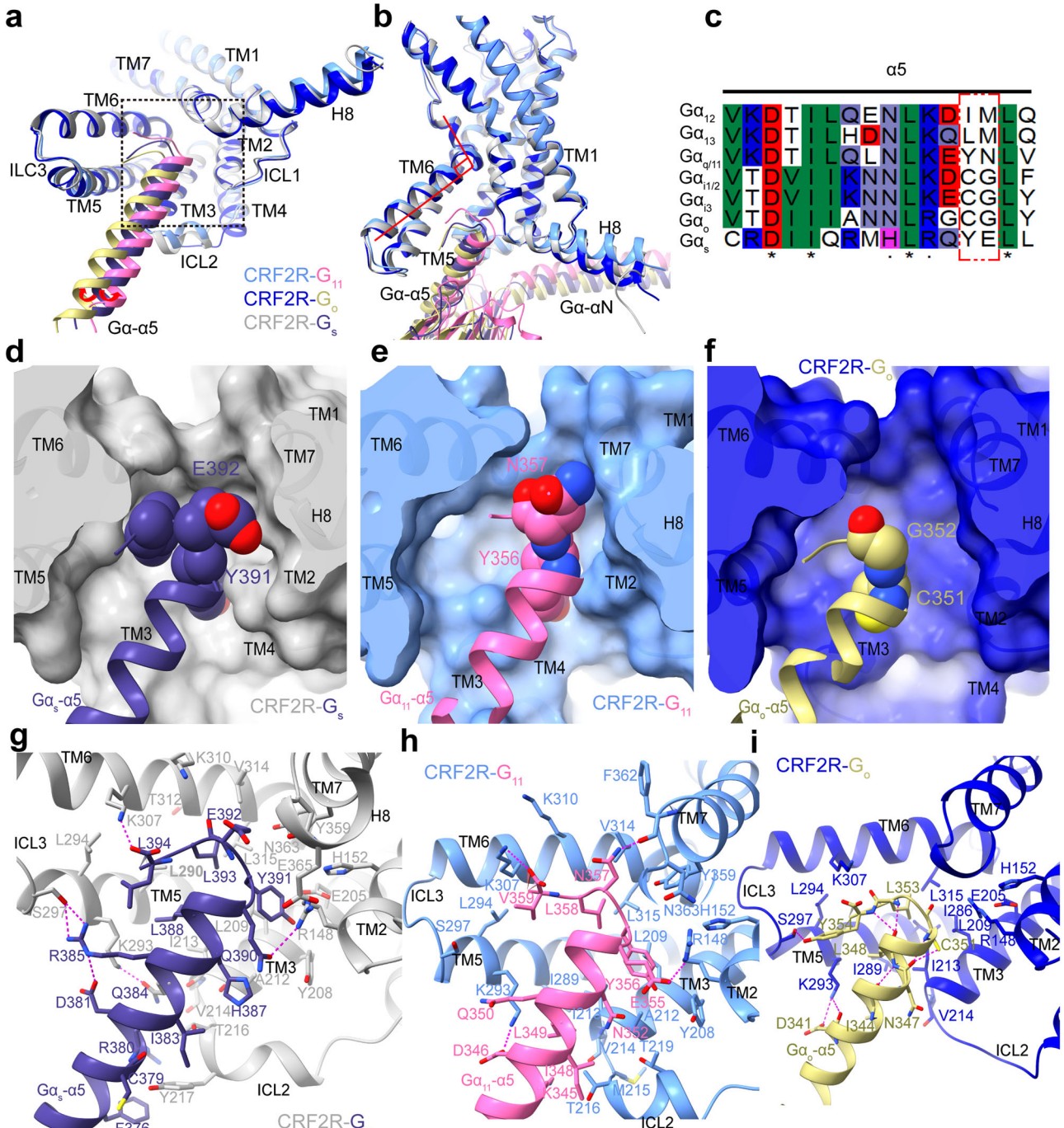

**Fig. 3 | Interaction patterns for the α5 helix of different G proteins.**
**a** Comparison of the TMD conformation and the position of the Gα-α5 helix C terminus in the G protein-bound structures is shown in cartoon representation in an intracellular view. The red arrows indicate the α5 helix of $G_{11}$ and $G_o$ shift away from TM5 or ICL2, compared to those of $G_s$. Structure of each CRF2R-G complex was superposed onto CRF2R-$G_{11}$ based on the receptor component. CRF2R (cornflower blue)-$G_{11}$ (hot pink), CRF2R (medium blue)-$G_o$ (dark khaki), CRF2R (dark gray)-$G_s$ (dark slate blue) (PDB: 6PB1). **b** Comparison of the TM6 conformation in three complexes. **c** Sequence alignment of α5 in the different Gα proteins. The red

box indicate the C-tail difference of G protein. **d**–**f** Binding pocket for the Gα-α5 C terminus. **d** UCN1-CRF2R-$G_s$ (PDB: 6PB1); **e** UCN1-CRF2R-$G_{11}$; **f** UCN1-CRF2R-$G_o$. The receptors are shown in cartoon and surface representations in an intracellular view. **g** Interactions between CRF2R and Gα-α5 in UCN1-CRF2R-$G_s$ (PDB: 6PB1).
**h** Interactions between CRF2R and Gα-α5 in UCN1-CRF2R-$G_{11}$. Many receptor side chains and those of K345, D346, and E355 on Gα-α5 were truncated in the structure, whose rotamers shown in this panel were from the Rosetta-refined model.
**i** Interactions between CRF2R and Gα-α5 in UCN1-CRF2R-$G_o$. The polar contacts are shown as purple dashed lines.

additional interface that seems a compensation for the smaller G protein binding interface with the shorter loop of ICL2 (Fig. 4a, e).

In the $G_s$-bound structure, ICL2 forms extensive interactions with αN, the β1 strand and the α5 helix. The receptor residues T216$^{ICL2}$, Y217$^{ICL2}$, E220$^{ICL2}$ and R223$^{4.41b}$ form a polar interaction network with the αN helix, the β1 strand and the α5 helix. Y217$^{ICL2}$ at the middle of this

helix, which is conserved in the CRF receptor family, inserts into a cavity formed by the N-terminal helix and the α5 helix of $G_s$ (Fig. 4c). Mutagenesis of Y217$^{ICL2}$ to alanine in CRF2R almost abolished coupling of $G_s$ (Fig. 4f, Supplementary Fig. 6a, d and Table 2). Furthermore, mutations in ICL2 residues V214A$^{3.59b}$, T216A$^{ICL2}$, S218A$^{ICL2}$, E220A$^{ICL2}$, R221A$^{ICL2}$, and L222A$^{ICL2}$ reduced $E_{max}$ in $G_s$ activation (Fig. 4f,

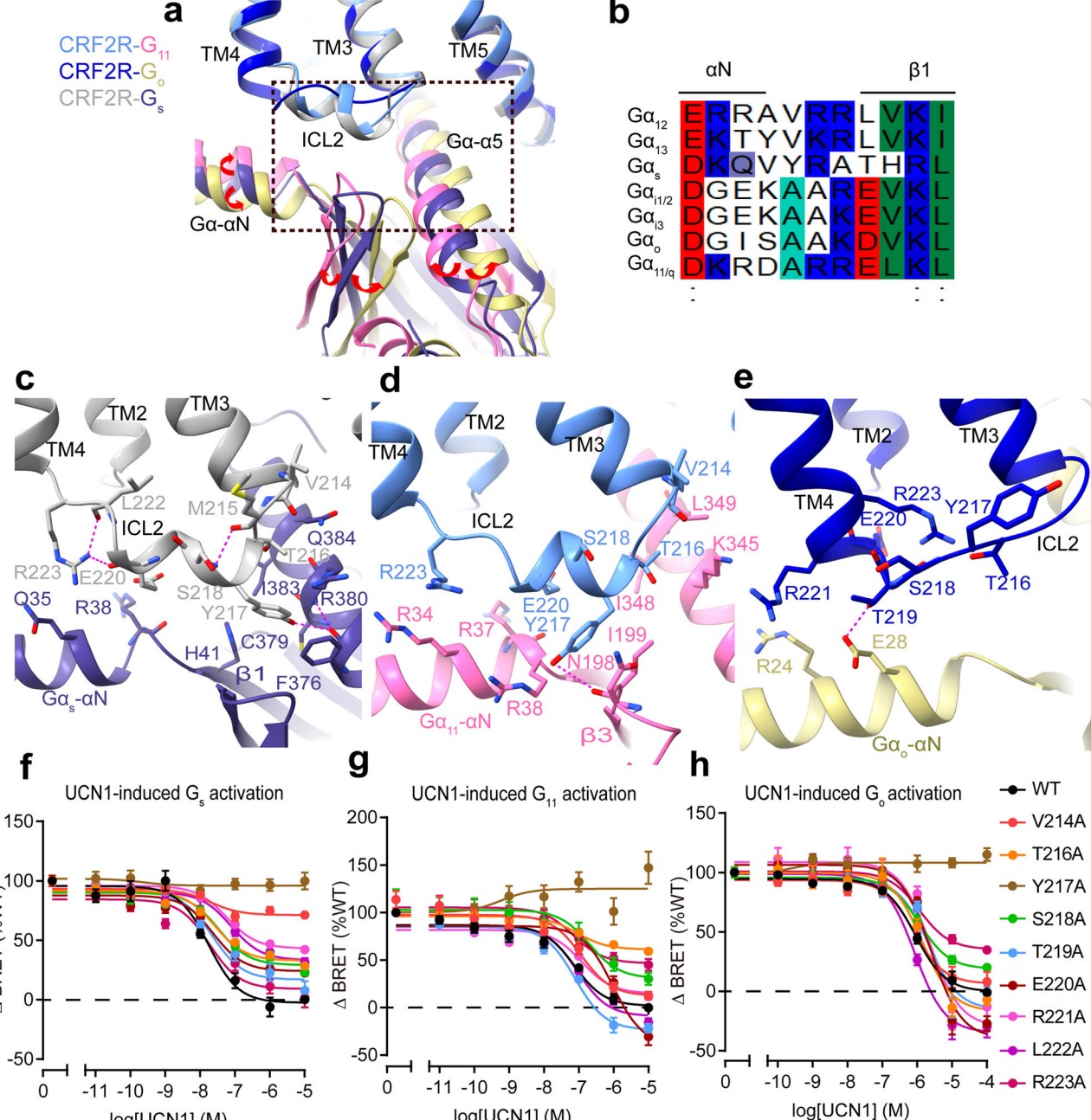

**Fig. 4 | G protein–binding interface mediated by the ILC2 of CRF2R.**
**a** Comparison of ICL2 conformations in the UCN1-CRF2R-$G_{11}$, UCN1-CRF2R-$G_o$ and UCN1-CRF2R-$G_s$ structures (PDB: 6PB1). The red arrows indicate the relative orientation differences of the different Gα. **b** Sequence alignment of αN and β1 in the different Gα proteins. **c** Interactions between ICL2 and $G_s$. **d** Interactions between ICL2 and $G_{11}$. The side chains of receptor residues, and N198, I199, and K345 of $G_{11}$ in this panel were truncated in the structure. Those residues shown here were prepared based on the Rosetta-refined model. **e** Interactions between ICL2 and $G_o$. The polar contacts are shown as purple dashed lines. **f** G protein activation and signaling assays of wild-type (WT) and ICL2 mutant CRF2R using a $Gα_s$-Gβγ dissociation assay **g** using a $Gα_{11}$-Gβγ dissociation assay and **h** using a $Gα_o$-Gβγ dissociation assay. Data from three independent experiments ($n = 3$) are presented as mean ± SEM.

Supplementary Fig. 6a, d and Table 2). In the $G_{11}$–bound structure, T216$^{ICL2}$, Y217$^{ICL2}$, E220$^{ICL2}$, R223$^{4.41b}$ formed an interface with the αN helix, the β2-β3 loop and the α5 helix of $G_{11}$. Y217$^{ICL2}$ formed hydrogen bonds with N198 (Fig. 4d, Supplementary Fig. 4e), mutation Y217$^{ICL2}$A also abolished coupling of $G_{11}$ (Fig. 4g, Supplementary Fig. 6b and Table 2). Besides Y217$^{ICL2}$, mutations in ICL2 residues T216$^{ICL2}$A, R223$^{4.41b}$A, also reduced $E_{max}$ in $G_{11}$ activation. Alanine substitution of E220$^{ICL2}$ showed clearly a great reduction in the potency of UCN1-mediated $G_{11}$ activation, a less degree but significant reduction in the potency of UCN1-mediated $G_o$ activation and almost no effect on the

potency of $G_s$ activation (Fig.4f–h, Supplementary Fig. 6a–c and Table 2), suggesting their critical roles in $G_{11}$ protein engagement and specificity. Similar interface of ICL2 with $G_{11}$ are also observed in the M1R-$G_{11}$ structure (Fig. 5b)[37]. In all these cases, the receptor ICL2 mediates G protein recognition, which may serve as a major determinant of G protein specificity.

In contrast, in CRFR-$G_o$ structure, ICL2 of CRF2R adopts an extended loop conformation, which is in a greater distance to the G protein and makes only limited contact with R24 and E28 in the αN helix of $G_o$ (Fig. 4e). Mutations of T216$^{ICL2}$, Y217$^{ICL2}$, S218$^{ICL2}$ and

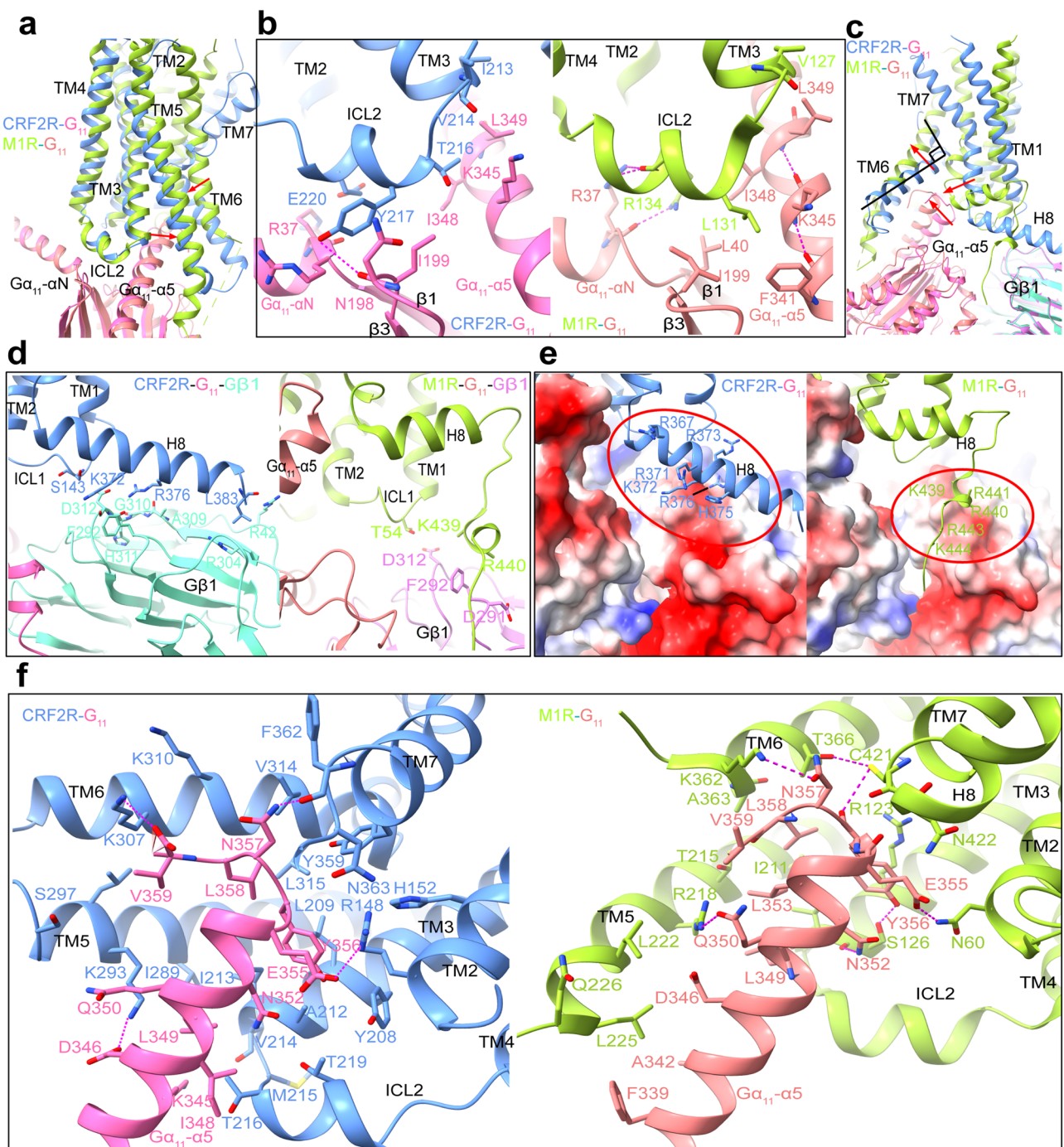

**Fig. 5 | Side-by-Side structure Comparison of CRF2R-G11 with M1R-G11.**
**a** Superposition of CRF2R-G11 and M1R-G11 (PDB: 6OIJ) complexes. The UCN1-CRF2R-G11 structure is colored cornflower blue (CRF2R), hot pink (G11) and aquamarine (Gα-β1); the M1R-G11 structure is colored green yellow (M1R), light coral (G11) and violet (Gα-β1). **b** Interaction comparison between ICL2 and G11 in CRF2R-G11 and M1R-G11 structures. Most ICL2 side chains of the receptor, the side chains of N198, I199, and K345 of Gα11 were truncated in the structure. The rotamers of those residues shown in this figure were based on the Rosetta-refined model. **c** Comparison of TM6, TM7 and H8-Gβ1 conformation in CRF2R-G11 and M1R-G11 complex. **d** Interactions comparison between CRF2R H8-Gβ1 and M1R H8-Gβ1. The

side chains of K372, and D379 of the receptor, and R42 and D312 of Gβ1were truncated in the structure, whose rotamers shown in this panel was from the Rosetta-refined model. **e** Comparison of positively charged residues in H8 of CRF2R, at the C-terminus of M1R and electrostatic surface potential of G protein. **f** Comparison of the Gα11-α5-TMD interactions in the CRF2R-G11 and M1R-G11 structures. Hydrogen bonds are shown as purple dashed lines. Most side chains of the receptor, and those of K345, D346 and E355 of Gα11 were truncated in the structure. The rotamers of those residues shown in this figure were based on the Rosetta-refined model.

R223A$^{4.41b}$ reduced $E_{max}$ in G$_o$ activation (Fig. 4h, Supplementary Fig. 6c and Table 2). R223$^{4.41b}$ of the receptor has distinct role for the interaction with different G proteins. In the receptor that couples G$_s$ or G11, R223$^{4.41b}$ is the second residue of a two-residue linker between

ICL2 helix and TM4, and is at the interface with the G protein. In the G$_o$-coupled receptor, however, R223$^{4.41b}$ is located at the second turn of TM4, with its side chain forming polar interactions with ICL2 residues T216$^{ICL2}$ and S218$^{ICL2}$ and stabilize the ICL2 loop

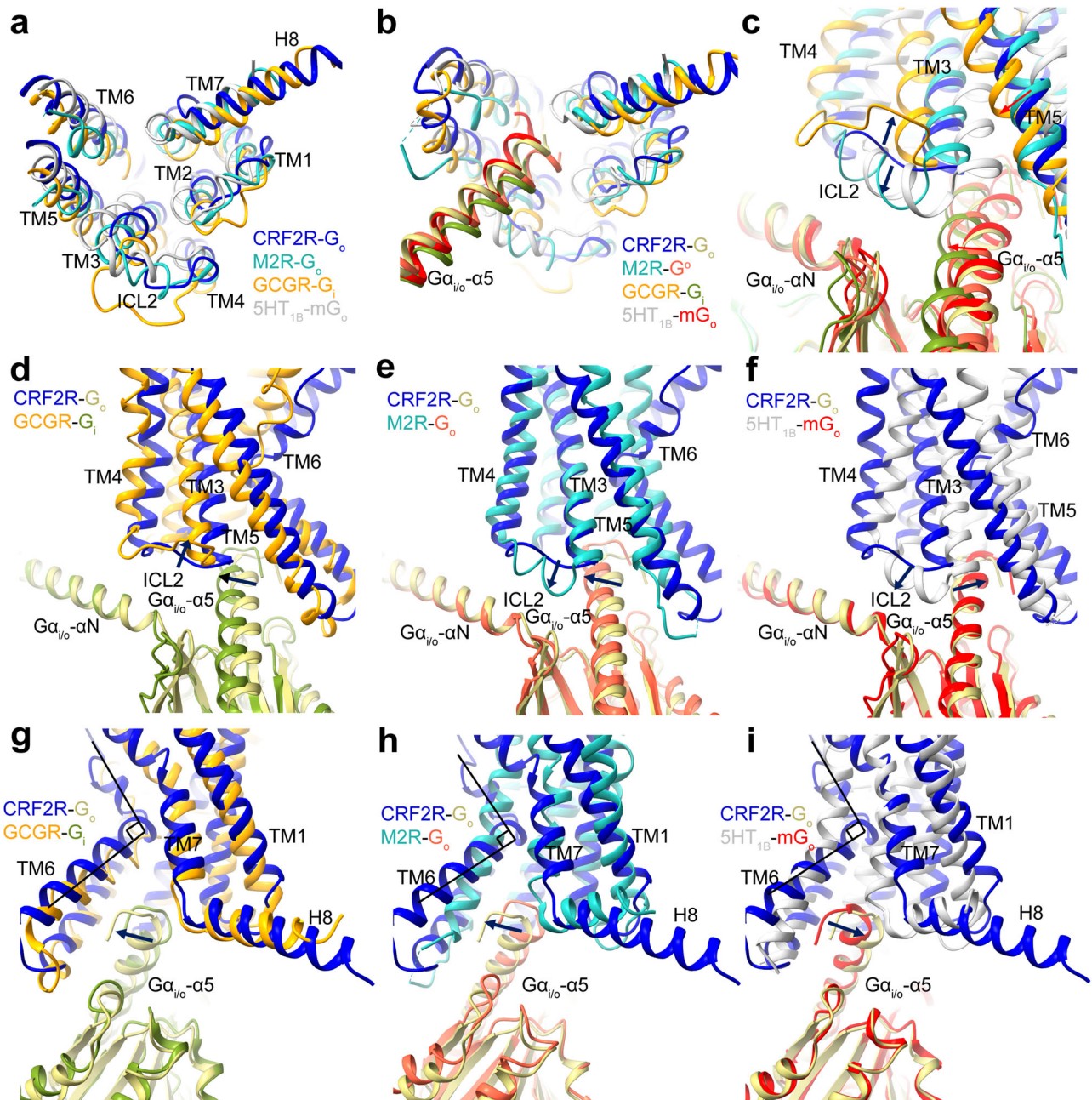

**Fig. 6 | Comparison of the Go and Gi bound GPCR structures. a–c** Comparison of the TMD conformation, the position of the Gα-α5 helix C terminus and the ICL2 conformation in the $G_o$ and $G_i$-bound GPCR structures. The CRF2R-$G_o$ structure is colored medium blue (CRF2R) and dark khaki ($G_o$); the M2R-$G_o$ structure is colored light sea green (M2R) and tomato ($G_o$) (PDB: 6OIK); the GCGR-$G_i$ structure is colored orange (GCGR) and olive drab ($G_i$) (PDB: 6LML); the 5HT$_{1B}$-mG$_o$ structure is colored silver (5HT$_{1B}$) and red (mG$_o$) (PDB: 6G79). **d** Comparison of the ICL2 conformational change in the CRF2R-$G_o$ and GCGR-$G_i$ structures. **e** Comparison of the ICL2 and Gα-α1 conformational change in the CRF2R-$G_o$ and M2R-$G_o$ structures. **f** Comparison of the ICL2 and Gα-α1 conformational change in the CRF2R-$G_o$ and 5HT$_{1B}$-mG$_o$ structures. **g** Comparison of the Gα-α5-TM6 interactions and the H8 conformational change in the CRF2R-$G_o$ and GCGR-$G_i$ structures. **h** Comparison of Gα-α5-TM6 interactions and the H8 conformation change in the CRF2R-$G_o$ and M2R-$G_o$ structures. **i** Comparison of Gα-α5-TM6 interactions and the H8 conformation change in the CRF2R-$G_o$ and 5HT$_{1B}$-mG$_o$ structures.

conformation (Fig. 4c–e). In addition, even though Y217$^{ICL2}$ does not directly interact with the hinge region of the $G_o$ protein and is in a greater distance to the $G_o$ protein than those in $G_{11}$ and $G_s$ complexes, Y217$^{ICL2}$ forms many interactions with surrounding residues that stabilize the ICL2 conformation in the $G_o$ complex. The Y217$^{ICL2}$A mutation likely destabilizes the ICL2 conformation, thus indirectly affect its coupling ability to $G_o$ (Fig. 4e, h, Supplementary Fig. 6c and Table 2).

The limited contacts between ICL2 and $G_o$ in the $G_o$-bound CRF2R structure most likely explain the lower potencies of UCN1 in

stimulating $G_o$ activation and signaling when compared to those in $G_s$- and $G_{11}$-coupled structures. A similar ICL2 loop conformation was observed in GCGR-$G_i$ structure[41], another class B GPCR that couples to $G_i$. The ICL2 loop of GCGR was located away from the $G_i$ protein and its interface with $G_i$ was weak (Fig. 6c, d). By contrast, $G_o$-coupled structures of class A GPCRs, including M2R and 5HT$_{1B}$, showed helical ICL2s closely interacted with the αN helix, β2-β3 loop and the α5 helix of $G_o$ (Fig. 6c, e, f)[37,50]. The above observations indicate that ICL2 is important for the G protein specificity of GPCRs.

## Molecular Recognition of the α5 helices of $G_s$, $G_{11}$ and $G_o$

Although all three active CRF2R-G complex structures are stabilized by extensive hydrophobic and polar interactions with Gα and Gβ of the G proteins, they show different molecular details in the recognition of the interactions from receptor to α5 helices of $G_s$, $G_{11}$ and $G_o$ (Fig. 3g–i). Comparing the recognition patterns for the α5 helix, we found that the bulky C-terminal α5 helix of $G_s$ from Q390 to L394 forms more extensive polar and hydrophobic interactions with the receptor than $G_{11}$ and $G_o$. Specifically, Y391 in the C-terminus of the $G_s$ α subunit, a key interface residue binds to a sub-pocket formed by R148$^{2.46b}$, H152$^{2.50b}$ and E205$^{3.50b}$, Y208$^{3.53b}$, L209$^{3.54b}$ of CRF2R. Other interface residues are i) $G_s$ E392, which forms polar contacts with K310$^{6.40b}$ and N363$^{8.47b}$ of the receptor; ii) Q390 of Gs, which forms a hydrogen bond with the side-chain of R148$^{2.46b}$ and iii) the C-terminal L394 of $G_s$, which forms a charge interaction with K307$^{6.37b}$ of the receptor TM6. In addition, Q384 and R385, at the middle of $G_s$-α5, forms hydrogen bond interactions with K293$^{5.64b}$ at the C-terminus of the receptor TM5 and S297$^{ICL3}$, respectively (Fig. 3g). Overall, the $Gα_s$-α5 helix extensively interacts with TM2, TM3, TM5, TM6, ICL2 and ICL3, and the TM7-H8 junction of the receptor.

Compared to that in the structure of $G_s$-coupled CRF2R, the C-terminal α5 helix of $G_{11}$ forms relatively few polar and hydrophobic contacts with the interface of CRF2R. Residues D346, L349, E355, Y356 and N357 on the α5 helix of $G_{11}$ are very important for the interaction with both CRF2R and M1R (Fig. 5f, Supplementary Fig. 4e). They form large polar and hydrophobic interaction networks with the receptor. A hydrogen bond interaction is observed between CRF2R TM5 residue K293$^{5.64b}$ and D346 on α5 of $G_{11}$, which is conserved in the interactions with $G_s$ (corresponding residue D381) and $G_o$ (corresponding residue D341). The carboxyl group of the C-terminal V359 of $G_{11}$-α5 is also observed to form a hydrogen bond with receptor residues K307$^{6.37b}$ and weak K310$^{6.40b}$. In addition, E355 and N357 of $G_{11}$-α5, forms an electrostatic interaction with R148$^{2.46b}$ and a hydrogen bond contact with F362$^{7.60b}$, respectively (Fig. 3h, Supplementary Fig. 4e), all are important for stabilizing the interface of α5 of $G_{11}$ with TM2, TM3, TM5, TM6 and TM7 of the receptor.

In contrast, the C-terminal helix of $G_o$ forms the fewest interactions with the receptor. D341 of $G_o$-α5, corresponding to D346 of $G_{11}$, forms hydrogen bonds interactions with K293$^{5.64b}$. The C-terminal residue Y354 of $G_o$ shows interactions with residues K293$^{5.64b}$ and forms a hydrogen bond interaction with the side chain of S297$^{ICL3}$ of the receptor (Fig. 3i). Mutation of S297$^{ICL3}$A completely abolished UCN1 potency on $G_o$ signaling (Supplementary Figs. 5h, 6c and Table 2). C351 of $G_o$-α5, corresponding to Y391 of $G_s$, has no bulky hydrophobic side chain. C351, therefore, forms only weak contacts with L209$^{3.54b}$ and I213$^{3.58b}$ on TM3 of the receptor. This is notably different to Y391 in $G_s$ that needs a larger sub-pocket in the receptor TM bundle. L353 of $G_o$-α5 interacts with the hydrophobic residues I213$^{3.58b}$, I286$^{5.57b}$, I289$^{5.60b}$, L315$^{6.45b}$ from TM3, TM5 and TM6, respectively (Fig. 3i). These hydrophobic interactions are very important for $G_o$ activation, and their key roles in $G_o$ activation were confirmed by our mutagenesis studies. All of alanine mutations of these hydrophobic residues greatly diminished $G_o$ activation. Specially, L315$^{6.45b}$A completely abolished UCN1 potency on $G_o$ signaling (Supplementary Fig. 5h, i, 6c and Table 2).

To study these important G protein subtype-specific interactions with the receptor (Supplementary Fig. 6e), we assessed UCN1-induced G protein subtype activation by serially mutated CRF2R using G protein dissociation assay and UCN1-induced cAMP accumulation assay (Supplementary Figs. 5, 6 and Table 2). Although the recognition patterns of the C-terminal α5 helix of Gα by the receptor is different, mutations of R148$^{2.46b}$A, H152$^{2.50b}$A, E205$^{3.50b}$A, L209$^{3.54b}$A, K293$^{5.64b}$A, L294$^{5.65b}$A, K307$^{6.37b}$A and L315$^{6.45b}$A, reduced UCN1 potency or $E_{max}$ in any G protein activation (Supplementary Figs. 5, 6 and Table 2). These alanine mutations break the interaction networks required in any G protein activation. I289$^{5.60b}$A abolished the coupling of $G_{11}$ and reduced

$E_{max}$ in $G_o$ activation, which only slightly alters $G_s$ activation. V314$^{6.44b}$A and Y359$^{6.45b}$A abolished the coupling of $G_{11}$, but reduced $E_{max}$ in $G_s$ activation, which only slightly alters $G_o$ activation. On the contrary, L290$^{5.60b}$A reduced efficiency and potency in $G_o$ activation, but showed significantly weaker effects on $G_s$ and $G_{11}$ activation. Receptor residue S297$^{ICL3}$ is at the interface with both Gα-α5 and β3 of any coupled G protein. Its mutation to alanine decreased UCN1 potency in any G protein activation and $G_o$ proteins appeared to be the most sensitive to amino acid substitution at S297$^{ICL3}$A because the C-terminal residue Y354 of $G_o$ forms a hydrogen bond with S297$^{ICL3}$, which is consistent with S301$^{ICL3}$ in CRF1R reported previously (Supplementary Fig. 5b, c, e, f, h, i, 6 and Table 2)[8]. Due to its interaction with specific G protein residues, this Ser plays a significant role in determining the G protein activation efficiency of the CRF receptors. Similar to the M1R-$G_{11}$ complex (PDB: 6OIJ), most of the interactions occurring between TM5 and TM6 of CRF2R and above identified $G_{11}$ residues may be critical for stabilizing the conformations of these TM segments for optimal interactions with the C terminus of the G protein (Fig. 5f).

## Discussion

Here, we show the structures of CRF2R-$G_{11}$ and CRF2R-$G_o$ complexes obtained by cryo-EM. Although the overall architecture of UCN1-CRF2R-$G_{11}$ and UCN1-CRF2R-$G_o$ complexes are similar to that of the CRF2R-$G_s$ complex, there are several universal and unique aspects of CRF2R coupling to different G protein subtypes. Firstly, there are clear conformational differences in ICL2 that may determine G protein recognition and specificity. Secondly, the sharp kink at the middle of TM6 facilitates formation of an open G protein–binding pocket, whose size may reflect the receptor's ability to couple to multiple G proteins. Class B GPCRs have a large pocket at the cytoplasmic surface of the receptor that can accommodate the relatively large size of the C termini of Gα subunits, providing the basis for the predominant coupling of class B GPCRs to $G_s$. Class B GPCRs couple less efficiently to $G_{q/11}$ and $G_{i/o}$. The ability of class B GPCRs to coupling G protein subtypes is $G_s$ > $G_{q/11}$ > $G_{i/o}$, which is consistent with the size of the C termini of Gα subunits, $G_s$ > $G_{q/11}$ > $G_{i/o}$. Therefore, the CRF receptors mainly couple to $G_s$ to mediate the cAMP-PKA pathway. It can also couple to $G_{q/11}$ to signal through the phospholipase C (PLC) pathway and exert multiple physiological actions[9,51]. For CRF2R, these signaling pathways regulate stress responses, blood pressure, food intake, and gastric emptying[7]. Although there are studies that implicate $G_o$ coupling of CRF2R[8], its physiological relevance remains unclear. However, it is possible that weak coupling interactions have physiological significance under certain circumstances[39]. Our structures and functional assays provide critical insights into the molecular mechanisms of class B GPCR activation through multiple G protein coupling and biased agonism through selective coupling of G protein subtypes.

## Methods

### Constructs of CRF2R and distinct classes of heterotrimeric G proteins

The human CRF2R (residues 2-388) was cloned into pFastBac vector. The native signal peptide was replaced with the haemagglutinin signal peptide (HA). To facilitate expression and purification, the LgBiT subunit (Promega) was fused via a 15 amino acid (GSSGGGGSGGGGSSG) linker (15aa) at the C terminus, followed by a TEV protease cleavage site and a double MBP (2MBP) tag to facilitate expression and purification. A dominant-negative human $Gα_{11}$ (DNGα$_{11}$) and $Gα_o$ (DNGαo) construct were generated based on the published DNGα$_s$[33]. Both DNGα$_{11}$ and DNGα$_o$ are chimera and the N termini of DNGα$_{11}$ and DNGα$_o$ were replaced with the N-terminus of Gα$_{i1}$, which can bind to scFv16[34]. In addition, we replaced the α-helical domain of Gα$_o$ (residues 59-175) with that of Gα$_{i1}$ (residues 59-174), which can bind Fab_G50 and introduced five mutations to create an Nb35 binding site. To facilitate the folding of the G protein, DNGα$_{11}$ was

coexpressed with GST-Ric-8A (gift from Dr. B. Kobilka) and DNGα$_o$ was coexpressed with GST-Ric-8B[52]. Rat Gβ1 with an N-terminal MHHHHHHSSGLVPRGSHMASHHHHHHHHHH-tag (His16) was fused with a SmBiT subunit (peptide 86, Promega)[53] via a 15 amino acid GSSGGGGSGGGGSSG linker at its C terminus.

In addition, to clone the constructs into the pBiT vector (Promega) for NanoBiT assays, constructs all contained an N-terminal FLAG tag (DYKDDDD) preceded by an HA signal sequence, and were cloned into the pcDNA3.1 vector (Invitrogen) for functional studies. All constructs were cloned using homologous recombination (Clone Express One Step Cloning Kit, Vazyme Biotech) and the primers were designed for site-direct mutagenesis studies (Supplementary Table 3).

## Expression and purification of CRF2R-G$_{11}$ and CRF2R-G$_o$ complexes

The CRF2R and G proteins were coexpressed in *Sf9* insect cells (Invitrogen). When the cells grew to a density of $3.0 \times 10^6$ cells per mL in ESF 921 cell culture medium (Expression Systems), we infected the cells with six separate virus preparations at a ratio of 1:3:3:3:3:3 for CRF2R-15aa-LgBiT-2MBP, DNG$_{11}$ or DNG$_o$, His16-Gβ1-peptide 86, Gγ2, scFv16 and GST-Ric-8A or GST-Ric-8B. The infected cells were cultured at 27 °C for 48 h before collection by centrifugation and the cell pellets were stored at −80 °C.

It was resuspended in 20 mM HEPES pH 7.4, 100 mM NaCl, 10 mM MgCl2, 10 mM CaCl2, 2 mM MnCl2, 10% glycerol, 0.1 mM TCEP, 25 mU/mL apyrase (Sigma), 10 μM UCN1(Synpeptide Co., Ltd),supplemented with Protease Inhibitor Cocktail (TargetMol, 1 mL/100 mL suspension). The lysate was incubated for 1 h at room temperature and complex from membranes solubilized by 0.5% (w/v) lauryl maltose neopentylglycol (LMNG, Anatrace) supplemented with 0.1% (w/v) cholesteryl hemisuccinate TRIS salt (CHS, Anatrace) for 2 h at 4 °C. The supernatant was isolated by centrifugation at $65,000 \times g$ for 40 min, and the solubilized complex was incubated with Amylose resin (NEB) for 2 h at 4 °C. The resin was loaded onto a plastic gravity flow column and washed with 15 column volumes of 20 mM HEPES, pH 7.4, 100 mM NaCl,10% glycerol,10 mM MgCl2, 1 mM MnCl2, 0.01% (w/v) LMNG, 0.01% glyco-diosgenin (GDN, Anatrace) and 0.004% (w/v) CHS, 2 μM UCN1, and 25 μM TCEP. After washing, the protein was treated overnight with TEV protease on column at 4 °C. Next day the flow through was collected and concentrated, then UCN1-CRF2R-G$_o$ and UCN1-CRF2R-G$_{11}$ were loaded onto a Superdex200 10/300 GL column and Superose6 Increase 10/300GL (GE Healthcare), respectively, with the buffer containing 20 mM HEPES, pH 7.4, 100 mM NaCl, 2 mM MgCl2, 0.00075% (w/v) LMNG, 0.00025% GDN, 0.0002% (w/v) CHS, 2 μM UCN1, and 100 μM TCEP. The complex fractions were collected and concentrated individually for electron microscopy experiments. The final yield of the purified complex was approximately 0.5 mg per liter of insect cell culture[21,33].

## Cryo-EM data acquisition

For the preparation of cryo-EM grids, 2.5 μL of the purified UCN1-CRF2R-G$_o$ and UCN1-CRF2R-G$_{11}$ complexes at a concentration of ~10.0 mg/ml were respectively applied to the glow-discharged Au 200 mesh and 300 mesh holey carbon grids (Quantifoil R1.2/1.3). The grids were blotted and then plunge-frozen in liquid ethane using a Vitrobot Mark IV (ThermoFisher Scientific).

Cryo-EM images of UCN1-CRF2R-G$_o$ were collected on a Titan Krios equipped with a Gatan K2 Summit direct electron detector in the Center of Cryo-EM, Zhejiang University. The microscope was operated at 300 kV accelerating voltage at a nominal magnification of 29,000 × in counting mode, corresponding to a pixel size of 1.014 Å. The total exposure time was set to 8 s with intermediate frames recorded every 0.2 s, resulting in an accumulated dose of 64 electrons per Å². A total of 2,929 movies were collected for the UCN1-CRF2R-G$_o$ complex.

Cryo-EM images of the UCN1-CRF2R-G$_{11}$ complex were collected on a Titan Krios equipped with a Gatan K3 Summit direct electron detector in Shanghai Institute of Materia Medica. The microscope was operated at 300 kV accelerating voltage, at a nominal magnification of 46,685× in counting mode, corresponding to a pixel size of 1.045 Å. In total, 4,122 movies were obtained with a defocus range of −1.2 to −2.2 μm. An accumulated dose of 80 electrons per Å² was fractionated into a movie stack of 36 frames.

## Image processing

Image stacks were subjected to beam-induced motion correction using MotionCor2.1[54]. Contrast transfer function (CTF) parameters for each micrograph were determined by Gctf v1.06[55]. The data processing was further performed in RELION-3.0[56].

For UCN1-CRF2R-G$_o$, auto-picking was performed by applying Laplacian-of-Gaussian blob detection and selected 1,840,659 particle projections that were subjected to reference-free 2D classification and averaging using a binned dataset with a pixel size of 2.028 Å. The subsets of 1,755,069 particle projections with well-defined averages were selected and subjected to 3D classification by employing a mask. One stable class accounting for 809,050 particles showed detailed features for all subunits and was subsequently subjected to further 3D classification with the alignment focusing on the complex. One subset showing high map quality with 171,435 particles was subject to CTF refinement, polishing, and 3D refinement. The final map has an indicated global resolution of 2.8 Å at a Fourier shell correlation (FSC) of 0.143. The local resolutions of this complex was determined using the Bsoft package (v.2.0.7) with half maps as input maps[57].

For UCN1-CRF2R-G$_{11}$, auto-picking was performed by applying Laplacian-of-Gaussian blob detection and selected 3,402,020 particle projections that were subjected to reference-free 2D classification and averaging using a binned dataset with a pixel size of 2.09 Å. The subsets of 2,825,032 particle projections with well-defined averages were selected and subjected to 3D classification. One good class accounting for 155,167 particles was subsequently subjected to 3D refinement. Further 3D classification with the alignment focusing on the complex by employing a mask, leading to the identification of the sub-dataset containing 94,765 particles. After last rounds of refinement, the final map has an indicated global resolution of 3.7 Å at a Fourier shell correlation (FSC) of 0.143. The local resolutions of this complex was determined using the Bsoft package (v.2.0.7) with half maps as input maps[57].

## Model building and refinement

The cryo-EM structure of the CRF2R-G$_s$-Nb35 complex (PDB code 6PB1) was used as the start for model building and refinement against the electron microscopy map. The model was docked into the electron microscopy density map using Chimera[58], followed by iterative manual adjustment and rebuilding in COOT[59]. Real space refinement using Phenix[35] were performed against the cryo-EM maps. Rosetta refinements[36] was performed for the UCN1-CRF2R-G$_{11}$ structure to further optimize the side chain rotamers. We used the Rosetta-refined model (including all Rosetta refined the side chains) for the purpose of discussion. We truncated many side chains in the Rosetta-refined model of the CRF2R-G$_{11}$ complex as the final model for Protein Data Bank deposit to reflect the quality of the electron density map. The model statistics were validated using MolProbity[60]. Fitting of the refined model to the final map was analysed using model-versus-map FSC. To monitor the potential over-fitting in model building, FSC$_{work}$ and FSC$_{free}$ were determined by refining 'shaken' models against unfiltered half-map-1 and calculating the FSC of the refined models against unfiltered half-map-1 and half-map-2. The final refinement statistics were provided in Supplementary Table 2. Structural figures were prepared in Chimera and PyMOL (https://pymol.org/2/).

## cAMP accumulation assay

The GloSensor cAMP assay was performed as previously described[61,62]. Briefly, HEK293 cells were transfected with the WT CRF2R or mutants and the GloSensor plasmid. 24 hours after transfection, cells were distributed into 96-well microplates at a density of $5 \times 10^4$ cells per well and incubated for another 24 hours at 37 °C in 5% $CO_2$. The cells were incubated with serum-free DMEM medium containing 2% GloSensor cAMP substrate (Promega) for 2 hours at 37 °C in 5% $CO_2$. The cells were then stimulated with increasing concentrations of UCN1. The luminescence was measured using an EnVision multi-label microplate detector (Perkin Elmer).

## G protein dissociation assay

The Gα dissociation from Gβγ assay was performed as previously described[63–65]. The plasmids WT or mutated CRF2R, Gα-RLuc8 (Gαs-RLuc8, Gα11-RLuc8 or Go-RLuc8), Gβ and Gγ-GFP were transiently co-transfected into HEK293 cells. The cells were re-seeded in 96-well microplates ($5 \times 10^4$ cells per well) and incubated for another 24 hours at 37 °C in 5% $CO_2$. The cells were washed twice with HBSS (Hank's Balanced Salt Solution) and stimulated with UCN1 at different concentrations for 2 min. The G protein dissociation signal was measured after the addition of the substrate coelenterazine 400a (5 μM) using a Mithras LB940 multimode reader (Berthold Technologies). The BRET signal was calculated as the ratio of light emission at 510 nm/400 nm.

## Measurement of receptor expression by ELISA assay

HEK293 cells were transiently transfected with N-terminal Flag-tagged wild type CRF2R or mutants in 24-well plates. 48 hours after transfection, the cells were fixed with 4% (w/v) formaldehyde for 10 min followed by incubation in blocking solution (5% BSA in DPBS) for 1 hour at room temperature. The cells were incubated with anti-FLAG primary antibody (Sigma Aldrich, Cat# F1804, 1:1000) followed by incubation with secondary anti-mouse antibody (Thermo Fisher, Cat# A-21235, 1:5000) conjugated to horseradish peroxide. The tetramethyl benzidine (TMB/E) solution was added and the reaction was terminated by adding 0.25 M HCl solution. The absorbance at 450 nm was measured using the TECAN luminescence counter (Infinite M200 Pro Nano-Quant) to characterize the cell surface expression level of each receptor. The expression levels of the mutants were normalized to that of the WT CRF2R. Data are shown as the mean ± SEM. Data are from three independent experiments ($n = 3$).

## Statistical analysis

All functional data were presented as means ± standard error of the mean (S.E.M.). Statistical analysis was performed using GraphPad Prism 8.0 (GraphPad Software). Experimental data were analyzed using two-sided one-way ANOVA with Tukey's test. $P < 0.05$ was considered statistically significant.

## Reporting summary

Further information on research design is available in the Nature Research Reporting Summary linked to this article.

## Data availability

Cryo-EM maps generated in this study have been deposited in the Electron Microscopy Data Bank under accession codes: EMD-26103 (G11-bound CRF2R receptor), EMD-26104 (Go-bound CRF2R receptor). The atomic coordinates generated in this study have been deposited in the Protein Data Bank under accession codes: 7TRY (G11-bound CRF2R receptor) and 7TS0 (Go-bound CRF2R receptor). Due to the limitation of the map resolution, many side chains of the G11-bound CRF2R structure were truncated before PDB deposition, as compared with our Rosetta-optimized model discussed herein. The structural model with Rosetta optimized side chains is provided as a Supplementary Data 1. Source data are provided with this paper.

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

## Acknowledgements

The cryo-EM data were collected at the Cryo-Electron Microscopy Research Center, the Shanghai Institute of Materia Medica and at the Center of Cryo-Electron Microscopy, Zhejiang University. We thanks for Z.Y. (Zhao Yang) provided the material and the new methods about the G protein dissociation. This work was supported by National Natural Science Foundation of China 32071203(L.H.Z.); the National Key R&D Program of China 2019YFA0904200 (J.P.S.); the Young Innovator Association of CAS 2018325(L.H.Z.) and SA-SIBS Scholarship Program to L.H.Z.; Ministry of Science and Technology (China) grants 2018YFA0507002 (H.E.X.), the Shanghai Municipal Science and Technology Major Project 2019SHZDZX02 (H.E.X.) and 18ZR1447800 (L.H.Z.); the CAS Strategic Priority Research Program XDB08020303 (H.E.X.); National Natural Science Foundation of China 31971195 (P.X.), 81922071 (Y.Z.) and 2100959(C.M.); Zhejiang Province Science Fund for Distinguished Young Scholars LR19H310001 (Y.Z.), Key R & D Projects of Zhejiang Province 2021C03039 (Y.Z.); Y.Z. is also supported by MOE Frontier Science Center for Brain Science & Brain-Machine Integration, Zhejiang University.

## Author contributions

L.H.Z. designed the expression constructs, purified the complexes, prepared the final samples for negative stain and data collection toward the structures, performed structure analysis, prepared figures and wrote the manuscript; L.H.Z., C.M., S.Y.J. and D.D.S. prepared the cryo-EM grids and collected cryo-EM images; L.H.Z., S.Y.J. and C.M. performed map calculations; S.Y.J., participated in figure preparation; L.H.Z. and X.E.Z. performed model building and structure analysis; X.E.Z. built and refined the structure models; X.H. performed measuring the interaction interface and the volume of bind pocket and interface; K.M. participated in manuscript writing; X.Y. designed the signaling experiment and supervised the project execution; J.P.S, P.X. and J.Y.L. conducted signaling experiments and data analysis; Y.Z. supervised C.M., D.D.S, S.Y.J, analyzed the structures, edited the manuscript. H.E.X. conceived the project, supervised L.H.Z., analyzed the data, and wrote the manuscript.

## Competing interests

The authors declare no competing interests.
