## [Peer Review File · Nature Communications]

Structure Insights into Selective Coupling of G Protein Subtypes by a Class B G Protein-Coupled ReceptorREVIEWER COMMENTS

Reviewer #1 (Remarks to the Author):

The manuscript “Structure Insights into Selective Coupling of G Protein Subtypes by a Class B G Protein-Coupled Receptor” by Zhao and coworkers targeted to determine high resolution cryo-EM structure of the CRF2R in complex with Go and G11 with the agonist UCN1 to investigate the molecular basis of G protein coupling specificity at the receptor. The Go and G11 complex structures were determined with an overall resolution of 2.8Å and 3.7Å, good enough to support the author’s conclusion. Based on the structures and in comparison to a previously determined structure of the CRF2R in complex with the canonical G protein Gs, the authors identified structural features that play an important role for the G protein coupling specificity of the receptor. These findings were further evaluated using an extensive set of mutational studies together with cellular signaling assays. Together, these analyses represent an important advancement in understanding G protein coupling specificity of class B GPCRs. Furthermore, this work presents the first class B GPCR structure in complex with a member of the Gq/11 family, which is an important milestone for the field. Nevertheless, I have some comments on the manuscript that should be addressed before publishing:

Major points:

- The authors describe differences in the relative orientation between the receptor and Gs, G11 and Go. They mention that the alpha5 helix of G11 and Go shift approx. 1.4 Å and 6.4Å away from ICL2. However, by looking at Fig. 3a, it seems like that the alpha5 helix of G11 is closer to the ICL2/TM4 region than Gs. Also the given distance change of 6.4Å reported for Go seems to be mostly based on the conformational change in ICL2 rather than movement of the C-terminal helix of the Galpha subunit. I would rather describe changes in the angle of alpha5 rather than distance changes relative to ICL2, which seems to undergo conformational changes as well.
- The authors claim that the preferential coupling of class B GPCRs to Gs can be explained by the open G protein binding pocket that is required to accommodate the bulky alpha5 helix of Gs. Since the authors mention that this binding pocket adapts to the size of the alpha5 C-terminal end, it is not entirely clear to me why the receptor should prefer a bulkier C-terminus over a smaller one. The authors should elaborate a little more on this to precise their view on the role of the size of the alpha5 helix on G protein coupling specificity. Especially under the observation that TM6 does not show any large difference between Gs, Gi and G11 coupled structures in class B receptors but seems to adapt its position in dependence of the G protein subtype for class A receptors.
- The authors discuss structural difference in the interface between the receptor and the different G protein subtypes. However, at least in the G11 structure the ICLs and the C-terminal end of the alpha5 helix seems to be poorly resolved regions. It would be worth trying some focused refinements to improve modeling of these regions in the G11 and Go complexes.

- The functional data for V214A is missing in Fig. 4f-h even though the authors mention it in the text (line 167)
- The authors mentioned that S218 interacts with the alphaN helix of G11. However, by looking at Fig. 4d, it seems to point towards the receptor core and should not interact with the G protein.
- Regarding the alanine mutagenesis experiments shown in Fig. 4: Alanine substitution of E220 clearly reduced the potency of UCN1 in the G11 activation assay, but I doubt that the effect of this mutation on the potency in the Go activation assay is statistically significant. Furthermore, the authors should comment on the fact that alanine mutation of Y217 also abolish Go coupling, even though it does not interact with the hinge region of the G protein and is in a greater distance to the G protein than for G11 and Gs.
- The authors mention that the potency of UCN1 for Go activation is lower when compared to Gs and G11. However, the authors compare G protein dissociation assays with an cAMP assay, which looks at a signal further downstream of G protein coupling and activation. This leads to signal amplification and can result in a left shift of the dose response curve. Therefore, it would be better to compare the UCN1 potency for Go and the canonical Gs protein in the same assay format (e.g. cAMP assay to analyze the Gs-dependent activation and Go-dependent inhibition of the adenylate cyclase).
- It would help to provide a schematic figure that shows the residues on the receptor and G protein side that are important for formation of the G protein subtype-specific interactions

Minor points:

Line 47: Please, cite Avet et al., 2022; Hauser et al., 2022 or Inoue et al., 2019. These papers have analyzed the promiscuity of GPCR in terms of G protein signaling.

Line 63: Please, change “coupled” to “couples”

Suppl. Fig. 2 and 3: The scale for the local resolution plot in Fig. S2c does not capture a useful dynamic range. A scale from 2.5 to 4.8 Å would be a better representation of the buildable density. A per-residue CC plot from the output of real-space refinement (Phenix) would be also useful for both models to see regions of lower model certainty, which is important for the interpretation of the results.

Line 99: I assume that the authors mean the alpha N helix and not the alpha1 helix of the Galpha subunit. The alpha1 helix is not the N-terminal helix, but is located between the beta1 sheet and the alpha helix of the AHD. This should also be changed in figures 2, 3, 4, 5, and 6, as well as in the text.

Fig. 2: I would be easier to understand the figure, if the name of the coupled G protein subtype would also be also listed under a) and e-i). I realize that the figure caption describes the color code and the corresponding G protein subtype, but it would be more convenient for the reader to find it in the figure directly.

Fig. 2c: I believe the residue 258 of CRF2R in the G11 complex should be labeled as a lysine and not arginine.

Lane 113: The comparison between the CRF2R-G11 and M1R-G11 complex needs some introduction. Otherwise, it is not really clear why those two receptor complexes were compared with each other. It should be highlighted that the structure of the M1R complex is the only G11 complex currently available besides the presented CRF2R-G11 structure.

Line 117: Please, replace “action” with “interaction”

Line 125: Please, change “Fig. 3a-c” to “Fig. 3a,b” – the panel c does not show information on the different interaction of the alpha5 helix with the receptor.

Line 132/Fig. 3d-f: Based on this figure it is hard to appreciate the described differences in the G protein volume. It would be better to show a clipped surface of the receptor together with the alpha5 helix, similar to Fig. 2e.

Line 184: Please, change to “another class B GPCR that couples to Gi...”

Line 186: Please, check this sentence.

Line 193: the alphaN helix of the G protein is not shown in Fig. 3g-i. Please, refer to an additional figure or remove alphaN at this position.

Figure 3g-i: Please, show sticks for the carboxyl group at the very c-terminus of alpha5 to better see its interaction with the receptor. Furthermore, I289 should be shown, since it abolishes the coupling of G11 and reduced the Bmax in Go activation. Please, also check if all the H-bonds mentioned in the text are highlighted with dashed lines (e.g. the H bond between S297-Y354 is not shown, although discussed in the text)

Figure 3h: Please, add the side chain of L294.

Figure 5f: It would be better for comparison, if the alpha5 helices of G11 bound to CRF2 and M1 would show the same orientation.

Reviewer #2 (Remarks to the Author):

Zhao et al. present a manuscript regarding G protein specificity of Class B GPCRs using as model system the structures of CRF2R coupled to G11 and Go. They compare these structures to the previously published structure of the same receptor bound to Gs. The authors have then the same receptor bound to the same agonist (UCN1) and coupled to different G proteins, which serves as the perfect platform to understand G protein coupling specificity and promiscuity. The authors perform cellular functional assays using BRET2 and cAMP assays to understand the structure to function relationships. The manuscript is well presented and highlights important features in G protein coupling selectivity.

However some of the methodology is slightly short on details. I recommend the manuscript for publication after some adjustments:

- Although the paper stands alone with the current data it would be greatly enhanced by looking at whether the differences found in the UNC1 binding site correlate with G protein coupling, i.e. finding a link between ligand binding and G protein coupling. This could be achieved by mutating residues at the receptor that contribute only to Gq/Go and not to Gs.

Additionally here are some minor comments regarding methodology:

- There is little detail about the use of Ric8 to aid in the assembly of G proteins for protein production. To my knowledge this is the first time that has been done for the production of GPCR-G protein complexes. The authors should provide more details on virus infection ratios or any other details they might find useful for reproducibility. Also it would be useful if authors could comment on the improvement achieved with this strategy since Go and G11 are generally produced in GPCR-G protein co-infections without many concerns.

- In cryo-EM preparation methods state that 200 mesh grids were used but then state in parenthesis 300 mesh, please clarify.

- Autopicking means with Laplacian, Gaussian? Please clarify.

- When talking about 3D classification in cryo-EM data processing, in both datasets it is stated that "Alignment focusing on the complex" or "alignment on the receptor". Does this mean using a mask? Signal subtraction? Please clarify.

- It is unclear how the authors made sure that model refinement did not have overfitting. The text says "The extent of any model overfitting during refinement was measured by refining the final model against one of the half-maps and by comparing the resulting map versus model FSC curves with the two half-maps and the full model" This is not the standard FSCwork/FSCfree, if an alternative value is used please clarify further.

Reviewer #3 (Remarks to the Author):

The authors obtain structures of CRF2R in complex with Go and G11 and compare these with the published structure CRF2R in complex with Gs. This comparison should be of general interest to the

GPCR community. My major concern is the quality of maps, particularly for the 3.7Å CRF2R-G11 complex, are not good enough to model many of the side chains discussed in the manuscript. The map quality for the G11 complex isn't adequate to model most of the side chains and even some of the backbone in the N-terminal peptide binding domain. This is true to a lesser extent for the Go complex. While the authors do not focus on the N-terminus, many scientists who download the structures will not look at the map quality and assume that the structures are accurate. I believe the manuscript could be revised to be acceptable for publication particular with the ample functional data; however, the authors need to acknowledge the limitations of their maps and the models need to be revised to reflect the map quality by stubbing where side chains are not clearly resolved in the map.

There are a few examples where interpretations are made that might be influenced by the map quality.

Line 100. "the N terminus of UCN in UCN1-CRF2R-G11 and UCN1-CRF2R-Go structures is folded inward and forms a hydrogen bond K258ECL2 in CFR2R,"

The map quality is inadequate to confidently model side chain of K258 in the G11 complex (Fig. 2c). Mutagenesis could show that K258 is important for activation of G11 and Go, but not Gs.

Line 112. "Corresponding interactions between the H8 of CRF2R and Gβ subunit were not observed in the CRF2R-G11 structure"

This statement should be modified to include "possibly due to poor map quality".

The map quality is not good enough to accurately model many of the sidechains of H8 shown in fig 2g.

Line 118. "In addition, there are some conformational changes of the Gα-α1 and Gα-α5 helices relative to the receptor"

Do you mean ..."Gα-α1 and Gα-αN"...?

Line 169. "Similar to the Gs-bound structure, T216ICL2, Y217ICL2, S218ICL2, E220ICL2 in the G11-bound structure also formed a polar interaction network with the αN helix, the β1-β2 loop and the α5 helix.

According to the model provided, T216 and S218 do not appear to form polar interactions with α N or α 5 helices, or the β 1- β 2 loop. Moreover, the quality of the maps in this region are not good enough to model sidechains with confidence. This should be acknowledged.

Line 171."S218|CL2 formed hydrophobic interactions with G11 (Fig. 4d)."

S218 points away from G11 in the model provided.

Minor point. The number of significant figures shown for data in tables is likely too high in many cases.

Point-by-Point Responses to the Comments Made by the Reviewers

Reviewer #1 (Remarks to the Author):

The manuscript “Structure Insights into Selective Coupling of G Protein Subtypes by a Class B G Protein-Coupled Receptor” by Zhao and coworkers targeted to determine high resolution cryo-EM structure of the CRF2R in complex with G_o and G₁₁ with the agonist UCN1 to investigate the molecular basis of G protein coupling specificity at the receptor. The G_o and G₁₁ complex structures were determined with an overall resolution of 2.8Å and 3.7Å, good enough to support the author’s conclusion. Based on the structures and in comparison to a previously determined structure of the CRF2R in complex with the canonical G protein G_s, the authors identified structural features that play an important role for the G protein coupling specificity of the receptor. These findings were further evaluated using an extensive set of mutational studies together with cellular signaling assays. Together, these analyses represent an important advancement in understanding G protein coupling specificity of class B GPCRs. Furthermore, this work presents the first class B GPCR structure in complex with a member of the G_{q/11} family, which is an important milestone for the field. Nevertheless, I have some comments on the manuscript that should be addressed before publishing:

Response: We thanks the reviewer for the positive comment on the importance of this work and the quality of the manuscript.

Major points:

-The authors describe differences in the relative orientation between the receptor and G_s, G₁₁ and G_o. They mention that the alpha5 helix of G₁₁ and G_o shift approx. 1.4 Å and 6.4Å away from ICL2. However, by looking at Fig. 3a, it seems like that the alpha5 helix of G₁₁ is closer to the ICL2/TM4 region than G_s. Also the given distance change of 6.4Å reported for G_o seems to be mostly based on the conformational change in ICL2 rather than movement of the C-terminal helix of the Galpha subunit. I would rather describe changes in the angle of alpha5 rather than distance changes relative to ICL2, which seems to undergo conformational changes as well.

Response: We totally agree with the above comments. In the revised manuscript, we have put our focus on the changes in the angle of alpha5 rather than the distance changes relative to ICL2.

- The authors claim that the preferential coupling of class B GPCRs to G_s can be explained by the open G protein binding pocket that is required to accommodate the bulky alpha5 helix of G_s. Since the authors mention that this binding pocket adapts to the size of the alpha5 C-terminal end, it is not entirely clear to me why the receptor should prefer a bulkier C-terminus over a smaller one. The authors should elaborate a little more on this to precise their view on the role of the size of the alpha5 helix on G protein coupling specificity. Especially under the observation that TM6 does not show any large difference between G_s, G_{i/o} and G_{q/11} coupled structures in class B receptors but seems to adapt its position in dependence of the G protein subtype for class A receptors.

Response: We thank the reviewer for the insightful suggestion. The issue has been solved by measuring interaction interface between the G α - α 5 C terminus (residues GH5.16 to GH5.26) and CRF2R via PDBePISA (J. Mol. Biol., 2007, 372, 774–797). TM6 helices of class B GPCRs adopt nearly identical conformations and generate a common binding pocket for coupling G_s, G_{i/o} and G_{q/11} protein. In contrast, structures of class A GPCR–G protein complexes have revealed differential positions of TM6 (Fig.5c and Fig.6g-i). Although CRF2R couples to three G protein subtypes through the binding of common binding pocket of CRF2R to the G α subunits, the ability of coupling with different G protein subtypes is different. CRF2R primarily activates cAMP-PKA pathways via G_s coupling. This can be explained by the key role of the C-terminal α 5 helix of G α subunit in the coupling selectivity. The fourth- and the third-last residues

from the C-terminal $\alpha 5$ helix of $G\alpha$ subunit are Y391 and E392 in G_s , C351 and G352 in $G_{i/o}$, Y356 and N357 in $G_{q/11}$. The bulkier residues in G_s require a larger receptor pocket than $G_{i/o}$ and $G_{q/11}$ to accommodate their side chains (Fig.3d-f). The measured interaction interface formed between the $\alpha 5$ C-terminus (residues G.H5.16 to G.H5.26) and CRF2R is larger in G_s complex (792.7 \AA^2) than those in G_{11} (524.7 \AA^2) and G_o (447.7 \AA^2) complexes. The bulkier side chains of Y391 and E392 in G_s allow a larger interaction interface between CRF2R and the G_s - $\alpha 5$ C terminus. Therefore, class B GPCRs perform their physiological actions by primarily coupling to G_s . While larger G protein-binding pockets of class B GPCRs are required to accommodate the bulkier $\alpha 5$ helix of G_s , they can still bind to the less bulky $\alpha 5$ helices from $G_{q/11}$ and $G_{i/o}$. Our observation from this study further confirmed the importance of the $G\alpha$ C termini as the main G protein selectivity determinants.

- The authors discuss structural difference in the interface between the receptor and the different G protein subtypes. However, at least in the G11 structure the ICLs and the C-terminal end of the alpha5 helix seems to be poorly resolved regions. It would be worth trying some focused refinements to improve modeling of these regions in the G11 and G_o complexes.

Response: We agree that the model building of the CRF2R- G_{11} structure is challenging because of the limited resolution of the density map (3.7 \AA). Based on the reviewer's suggestion, we have further inspected both CRF2R- G_{11} and CRF2R- G_o models against the density maps, and have performed refinement with particular focus on the alpha5 helices and the IC loops of the models. Global refinement with Rosetta (in phenix package) has also been performed to optimize the side chain rotamers. Both structures show improved model-map correlation coefficients and molprobity scores (please see the revised supplemental table 2.)

- The functional data for V214A is missing in Fig. 4f-h even though the authors mention it in the text (line 167)

Response: This point has been taken and the functional data for V214A has been added in Fig.4f-h and the figures have been updated in our revised manuscript.

Fig. 4f-h (f) G protein activation and signaling assays of wild-type (WT) and ICL2 mutant CRF2R using a $G\alpha_s$ - $G\beta\gamma$ dissociation assay (g) using a $G\alpha_{11}$ - $G\beta\gamma$ dissociation assay and (h) using $G\alpha_o$ - $G\beta\gamma$ dissociation assay. Hydrogen bonds are shown as purple dashed lines.

- The authors mentioned that S218 interacts with the alpha N helix of G11. However, by looking at Fig. 4d, it seems to point towards the receptor core and should not interact with the G protein.

Response: We thank the reviewer for this comment. We have checked map density and S218 indeed

points away from G_{11} in the model and should not interact with the G_{11} protein. We have deleted this comment on “S218 interacts with the alpha N helix of G_{11} ”

- Regarding the alanine mutagenesis experiments shown in Fig. 4: Alanine substitution of E220 clearly reduced the potency of UCN1 in the G_{11} activation assay, but I doubt that the effect of this mutation on the potency in the G_o activation assay is statistically significant. Furthermore, the authors should comment on the fact that alanine mutation of Y217 also abolish G_o coupling, even though it does not interact with the hinge region of the G protein and is in a greater distance to the G protein than for G_{11} and G_s .

Response: We appreciate this comment. We have repeated these functional assays by three more times, double checked our raw data, and performed statistical analysis accordingly. Alanine substitution of E220^{ICL2} showed clearly a great reduction in the potency of UCN1-mediated G_{11} activation, a less degree but significant reduction in the potency of UCN1-mediated G_o activation and almost no effect on the potency of G_s activation (Fig.4f-h, Supplementary Fig. 6a-c, also see Supplementary table 2). We have revised our manuscript accordingly based on the above comments.

Regarding the alanine mutation of Y217, even though Y217^{ICL2} does not directly interact with the hinge region of the G_o protein and is in a greater distance to the G_o protein than those in G_{11} and G_s complexes, Y217^{ICL2} forms many interactions with surrounding residues that stabilize the ICL2 conformation in the G_o complex. The Y217A mutation likely destabilizes the ICL2 conformation, thus indirectly affect its coupling ability to G_o .

Supplementary Fig. 6a-c Bars represent differences in calculated potency of UCN1(pEC_{50} [half maximal effective concentration]) for each mutation relative to WT of CRF2R (left) and the maximal response (right) of wild-type and mutants CRF2R on the $G\alpha_s$ - $G\beta\gamma$ dissociation assay (a) $G\alpha_{11}$ - $G\beta\gamma$ dissociation assay (b) and $G\alpha_o$ - $G\beta\gamma$ dissociation assay (c) in HEK293 cells in response to UCN1 stimulation. The maximal responses of mutants CRF2R were normalized to that of WT CRF2R. Data are shown as the mean \pm SEM of at least three independent experiments ($n \geq 3$). Statistical differences between WT and mutants were determined by one-way ANOVA with Dunnett's test. * $p < 0.05$; ** $p < 0.01$; *** $p < 0.001$. ND, not detectable.

- The authors mention that the potency of UCN1 for G_o activation is lower when compared to G_s and G_{11} . However, the authors compare G protein dissociation assays with an cAMP assay, which looks at a signal further downstream of G protein coupling and activation. This leads to signal amplification and can result in a left shift of the dose response curve. Therefore, it would be better to compare the UCN1 potency for G_o and the canonical G_s protein in the same assay format (e.g. cAMP assay to analyze the G_s -dependent activation and G_o -dependent inhibition of the adenylate cyclase).

Response: We thank the reviewer for the helpful suggestions. To further confirm the effect of the mutations of CRF2R on cAMP accumulation, we performed the G_s dissociation assays, analogous to the G_o and G_{11} dissociation assays. The data shows that a similar pattern is observed for those mutations as shown in cAMP accumulation assay (Supplementary Fig. 5a-i, 6a-c, also see Supplementary table 2). Thus, we compared the UCN1 potency for G_s , G_o and G_{11} protein activation via a $G\alpha$ - $G\beta\gamma$ dissociation assay.

Supplementary Fig. 5a-i: G-protein activation of CRF2R mutants using a G α_s -G $\beta\gamma$ dissociation assay (a-c), a G α_{11} -G $\beta\gamma$ dissociation assay (d-f) and a G α_o -G $\beta\gamma$ dissociation assay (g-i).

- It would help to provide a schematic figure that shows the residues on the receptor and G protein side that are important for formation of the G protein subtype-specific interactions.

Response: We thank the reviewer for this insightful suggestion. We have added a schematic figure that shows the residues on the receptor and G protein sides that are important for the formation of the G protein subtype-specific interactions as below, which is also included as Supplementary Fig. 5j in the revised manuscript.

Supplementary Fig. 5j: (j) The important residues on CRF2R and G proteins for the formation of the G protein subtype-specific interactions are colored pink, and the yellow represents that alanine mutations completely abolished UCN1 potency on G protein signaling.

Minor points:

Line 47: Please, cite Avet et al., 2022; Hauser et al., 2022 or Inoue et al., 2019. These papers have analyzed the promiscuity of GPCR in terms of G protein signaling.

Response: We thank the reviewer for this valuable suggestion. We have modified the text to make clear the promiscuity of GPCRs in terms of G protein signaling and cited the paper of Avet et al., 2022; Hauser et al., 2022 or Inoue et al., 2019. (References11-13)

Line 63: Please, change “coupled” to “couples”

Response: We thank the reviewer for the careful reviewing. We have made a modification to the revised manuscript.

Suppl. Fig. 2 and 3: The scale for the local resolution plot in Fig. 2c does not capture a useful dynamic range. A scale from 2.5 to 4.8 Å would be a better representation of the buildable density. A per-residue CC plot from the output of real-space refinement (Phenix) would be also useful for both models to see regions of lower model certainty, which is important for the interpretation of the results.

Response: We thank the reviewer for the helpful suggestions and valuable comments. We have revised the scale for the local resolution plot accordingly. We also included the FSC curves and the per-residue CC plot from Phenix refinement for both models in the manuscript according to reviewer’s suggestion. The corresponding figure has been revised as shown below.

Supplementary Fig. 2c. (c) Cryo-EM map of the UCN1-CRF2R-G₁₁ complex, colored by local resolution (Å) calculated using the Bsoft package.

Supplementary Fig. 2b. (b) Flowchart of cryo-EM data analysis.

Supplementary Fig. 2e. e) per-residue CC plot of the CRF2R from the output of real-space refinement in Phenix 1.16.

Supplementary Fig. 3b. (b) Flowchart of cryo-EM data analysis.

Supplementary Fig. 3e. (e) per-residue CC plot of the CRF2R from the output of real-space refinement in Phenix 1.16.

Line 99: I assume that the authors mean the alpha N helix and not the alpha1 helix of the Galpha subunit. The alpha1 helix is not the N-terminal helix, but is located between the beta1 sheet and the alpha helix of the AHD. This should also be changed in figures 2, 3, 4, 5, and 6, as well as in the text.

Response: We thank the reviewer for the careful reviewing. We have distinguished the alpha N helix and the alpha1 helix of the Galpha subunit. We also have changed them in figures 2, 3, 4, 5, and 6, as well as in the main text.

Fig. 2: I would be easier to understand the figure, if the name of the coupled G protein subtype would also be also listed under a) and e-i). I realize that the figure caption describes the color code and the corresponding G protein subtype, but it would be more convenient for the reader to find it in the figure directly.

Response: We appreciate the reviewer for the insightful comments and constructive suggestions, which have helped us improve the quality of this manuscript and make the figures more convenient for readers to understand. We have listed the name of the coupled G protein subtypes under Fig. 2 a) and e-i) in the revised manuscript.

Fig. 2c: I believe the residue 258 of CRF2R in the G11 complex should be labeled as a lysine and not arginine.

Response: We thank the reviewer for the careful reading and we have made a modification in the revised

manuscript.

Line 113: The comparison between the CRF2R-G11 and M1R-G11 complex needs some introduction. Otherwise, it is not really clear why those two receptor complexes were compared with each other. It should be highlighted that the structure of the M1R complex is the only G11 complex currently available besides the presented CRF2R-G11 structure.

Response: We appreciate the reviewer's suggestion. We have added that M1R-G₁₁ is the only available G₁₁ binding GPCR structure, and the comparison of CRF2R-G₁₁ with M1R-G₁₁ complex has clearly shown the similarity and difference in structural features of the two G₁₁ complexes.

Line 117: Please, replace “action” with “interaction”

Response: It has been corrected.

Line 125: Please, change “Fig. 3a-c” to “Fig. 3a, b” – the panel c does not show information on the different interaction of the alpha5 helix with the receptor.

Response: It has been changed.

Line 132/Fig. 3d-f: Based on this figure it is hard to appreciate the described differences in the G protein volume. It would be better to show a clipped surface of the receptor together with the alpha5 helix, similar to Fig. 2e.

Response: We thank the reviewer for the insightful suggestion. The corresponding figure has been revised.

Line 184: Please, change to “another class B GPCR that couples to Gi...”

Response: We thank the reviewer for the careful reading. We have made a modification in the revised manuscript.

Line 186: Please, check this sentence.

Response: The sentence has been revised.

Line 193: the alphaN helix of the G protein is not shown in Fig. 3g-i. Please, refer to an additional figure or remove alphaN at this position.

Response: We thank the reviewer for the insightful suggestion and have removed alphaN at this position in the revised manuscript.

Figure 3g-i: Please, show sticks for the carboxyl group at the very c-terminus of alpha5 to better see its interaction with the receptor. Furthermore, I289 should be shown, since it abolishes the coupling of G11 and reduced the Bmax in Go activation. Please, also check if all the H-bonds mentioned in the text are highlighted with dashed lines (e.g. the H bond between S297-Y354 is not shown, although discussed in the text)

Response: We appreciate the reviewer for the careful reading and constructive suggestions. We have shown sticks for the carboxyl group at the very C-terminus of alpha 5 in Figure 3g-i. I289 have been shown in Figure 3h-i and we have highlighted H-bonds with dashed lines as mentioned in the main text, including the H bond between S297-Y354 in the revised manuscript.

Figure 3h: Please, add the side chain of L294.

Response: We thank the reviewer for the careful reading. This has been done in Fig. 3i and Fig. 3i has been replaced Fig. 3h in the revised manuscript. We have checked the map density and found that

L294^{5.65b} should not interact with the Y354 of G_o protein. We have deleted this comment on “The C-terminal residue Y354 of G_o shows hydrophobic interactions with L294^{5.65b} ” in the main text and in the figure.

Figure 5f: It would be better for comparison, if the alpha5 helices of G11 bound to CRF2 and M1 would show the same orientation.

Response: We thank the reviewer for the insightful suggestion and have revised the Fig. 5f.

Reviewer #2 (Remarks to the Author):

Zhao et al. present a manuscript regarding G protein specificity of Class B GPCRs using as model system the structures of CRF2R coupled to G11 and G_o. They compare these structures to the previously published structure of the same receptor bound to G_s. The authors have then the same receptor bound to the same agonist (UCN1) and coupled to different G proteins, which serves as the perfect platform to understand G protein coupling specificity and promiscuity. The authors perform cellular functional assays using BRET2 and cAMP assays to understand the structure to function relationships. The manuscript is well presented and highlights important features in G protein coupling selectivity. However some of the methodology is slightly short on details. I recommend the manuscript for publication after some adjustments:

Response: We thank the reviewer for the positive assessment on the quality and importance of our works reported in this paper. We would also like to thank the reviewer for the careful and thorough reading of this manuscript and for the thoughtful comments and constructive suggestions, which have helped us improve the quality of this manuscript. We have revised the manuscript as detailed below.

- Although the paper stands alone with the current data it would be greatly enhanced by looking at whether the differences found in the UCN1 binding site correlate with G protein coupling, i.e. finding a link between ligand binding and G protein coupling. This could be achieved by mutating residues at the receptor that contribute only to Gq/Go and not to G_s.

Response: We thank the reviewer for the insightful suggestions. Our experiments clearly show that some mutations at the TMD of the receptor affect only the coupling of G₁₁/G_o, but not to that of G_s. For example, K258^{ECL2}A reduced efficiency and potency in G₁₁ and G_o activation, but it showed no significant effects on G_s activation. I289^{5.60b}A abolished the coupling of G₁₁ and reduced potency in G_o activation, but only slightly alters G_s activation. V314^{6.44b}A and Y359^{6.45b}A abolished the coupling of G₁₁, but reduced E_{max} in G_s activation, which only slightly alters G_o activation. L290^{5.61b}A reduced efficiency and potency in G_o activation, but showed significantly weaker effects on G_s and G₁₁ activation. S297^{ICL3}A decreased UCN1 efficiency in any G protein activation and G_o proteins appeared to be the most sensitive to amino acid substitution at S297^{ICL3}A because the C-terminal residue Y354 of G_o forms a hydrogen bond with S297^{ICL3}, which is consistent with the effects of S301^{ICL3} mutation in CRF1R reported previously. The corresponding description could be found at line 103, line279, line280, line281, line284 and the data is shown in Supplementary Fig. 5b-c, e-f, h-i, 6 and table 2.

Additionally here are some minor comments regarding methodology:

- There is little detail about the use of Ric8 to aid in the assembly of G proteins for protein production. To my knowledge this is the first time that has been done for the production of GPCR-G protein complexes. The authors should provide more details on virus infection ratios or any other details they might find useful for reproducibility. Also it would be useful if authors could comment on the improvement achieved with this strategy since G_o and G11 are generally produced in GPCR-G protein co-infections without

many concerns.

Response: We thank the reviewer for the careful reading. Ric-8A and Ric-8B are nonreceptor G protein guanine nucleotide exchange factors that collectively bind the four subfamilies of G protein subunits. Co-expression of G α subunits with Ric-8A or Ric-8B in HEK293 cells or insect cells greatly promoted G α protein expression [Chan, P. *et al.* Purification of heterotrimeric G protein alpha subunits by GST-Ric-8 association: primary characterization of purified G alpha(olf). *The Journal of biological chemistry* 286, 2625-2635, doi:10.1074/jbc.M110.178897 (2011)]. We thus included Ric-8A for G₁₁ and Ric-8B for G_o to increase the expression yield. The detailed description has been included in the Material and Methods section of the revised manuscript.

- In cryo-EM preparation methods state that 200 mesh grids were used but then state in parenthesis 300 mesh, please clarify.

Response: We appreciate the reviewer for the careful reading of our manuscript. The sample of UCN1-CRF2R-G_o complex was applied to the glow-discharged Au 200 mesh holey carbon grids and UCN1-CRF2R-G₁₁ complex was applied to the glow-discharged Au 300 mesh holey carbon grids. We have clarified that in cryo-EM preparation methods section of the revised manuscript.

- Autopicking means with Laplacian, Gaussian? Please clarify.

Response: We appreciate the reviewer for pointing out the problem. Yes, we performed autopicking by applying Laplacian-of-Gaussian blob detection. In addition, we have added the information in the Materials and Methods section accordingly.

- When taking about 3D classification in cryo-EM data processing, in both datasets it is stated that “Alignment focusing on the complex” or “alignment on the receptor”. Does this mean using a mask? Signal subtraction? Please clarify.

Response: We thank the reviewer for the valuable considerations. We indeed performed 3D classification with the alignment focusing on the complex by employing a mask in both datasets. We now have clarified this point in the Materials and Methods section.

- It is unclear how the authors made sure that model refinement did not have overfitting. The text says “The extent of any model overfitting during refinement was measured by refining the final model against one of the half-maps and by comparing the resulting map versus model FSC curves with the two half-maps and the full model” This is not the standard FSC_{work}/FSC_{free}, if an alternative value is used please clarify further.

Response: We thank the reviewer for this important consideration. Fitting of the refined model to the final map was analyzed using model-versus-map FSC. To monitor the potential over-fitting in model building, FSC_{work} and FSC_{free} were determined by refining ‘shaken’ models against unfiltered half-map-1 and calculating the FSC of the refined models against unfiltered half-map-1 and half-map-2”. We have revised the manuscript accordingly by adding this information in the Materials and Methods section and we also have supplemented the figures in Supplementary Fig. 2 and Supplementary Fig. 3.

Reviewer #3 (Remarks to the Author):

The authors obtain structures of CRF2R in complex with G_o and G₁₁ and compare these with the published structure CRF2R in complex with G_s. This comparison should be of general interest to the GPCR community. My major concern is the quality of maps, particularly for the 3.7Å CRF2R-G₁₁ complex, are not good enough to model many of the side chains discussed in the manuscript. The map quality for the G₁₁ complex isn't adequate to model most of the side chains and even some of the

backbone in the N-terminal peptide binding domain. This is true to a lesser extent for the Go complex. While the authors do not focus on the N-terminus, many scientists who download the structures will not look at the map quality and assume that the structures are accurate. I believe the manuscript could be revised to be acceptable for publication particular with the ample functional data; however, the authors need to acknowledge the limitations of their maps and the models need to be revised to reflect the map quality by stubbing where side chains are not clearly resolved in the map.

Response: We thank the reviewer's comments and understand the reviewer's concerns about the map quality particularly of the G₁₁ structure. We have inspected the models and truncated many side chains that have no clear density, particularly those in extracellular domain of the CRF2R-G₁₁ complex structure. To optimize the side chain rotamers, we have performed Rosetta refinement (phenix.rosetta_refine) for both structures, in which the side chain rotamers were optimized based on not only map density but also the restraints of the chemical environment of the residues. The model of the N-terminal domain of CRF2R was originally adopted from a 2.5 Å crystal structure (3N93), which was docked nicely in the N-terminal region of the receptor of the density maps. The side chains of this region have also been optimized by rosetta refinement.

There are a few examples where interpretations are made that might be influenced by the map quality.

Line 100. "the N terminus of UCN in UCN1-CRF2R-G11 and UCN1-CRF2R-Go structures is folded inward and forms a hydrogen bond K258ECL2 in CFR2R,"

Response: We have carefully inspected the density maps of this region. While it is chemically reasonable, we decided not to build the first residue Asp of UCN1 because the weak density is not good enough to support the orientations of this residue. The sentence is therefore deleted in the revised manuscript. Fig.2 is corrected in revised manuscript.

The map quality is inadequate to confidently model side chain of K258 in the G11 complex (Fig. 2c). Mutagenesis could show that K258 is important for activation of G11 and Go, but not Gs.

Response: We thank the reviewer for the invaluable advice. We have examined the effect of mutation of K258A on the potency of CRF2R activation using G α -G $\beta\gamma$ dissociation assays (G α_s -G $\beta\gamma$, G α_{11} -G $\beta\gamma$ and G α_o -G $\beta\gamma$ dissociation assay). Our data has shown that K258 is important for activation of G₁₁ and G_o, but not G_s. Please see the data in Supplementary Fig. 5 b, e, h, 6a-c and Supplementary table 2.

Line 112. "Corresponding interactions between the H8 of CRF2R and G β subunit were not observed in the CRF2R-G11 structure". This statement should be modified to include "possibly due to poor map quality".

The map quality is not good enough to accurately model many of the sidechains of H8 shown in fig 2g. (Lihua)

Response: We agree with the reviewer and have made corresponding modification as the reviewer suggested.

Line 118. "In addition, there are some conformational changes of the G α - α 1 and G α - α 5 helices relative to the receptor"

Do you mean ..."G α - α 1 and G α - α N"...?

Response: We appreciate the reviewer for the comment and we have corrected the G α - α 1 to the G α - α N in the revised manuscript.

Line 169. "Similar to the G_s-bound structure, T216ICL2, Y217ICL2, S218ICL2, E220ICL2 in the G11-bound structure also formed a polar interaction network with the α N helix, the β 1- β 2 loop and the α 5 helix.

According to the model provided, T216 and S218 do not appear to form polar interactions with α N or α 5 helices, or the β 1- β 2 loop. Moreover, the quality of the maps in this region are not good enough to model sidechains with confidence. This should be acknowledged.

Response: We thank the reviewer for this comment. We have revised the main text accordingly. We agree that the interface between the ICL2 of the receptor and G₁₁ alpha subunit is formed largely by hydrophobic and/or van de Waals interactions among main chain atoms and/or the carbon backbones of the side chains, of the interface residues. Due to the limited map resolution, a few side chains in this region cannot be clearly defined by the density map alone. We performed rosetta refinement (phenix.rosetta_refine) which optimized the side chain rotamers based not only on electron density but also on restraints of chemical environment.

Line 171.”S218^{ICL2} formed hydrophobic interactions with G₁₁ (Fig. 4d).”

S218 points away from G₁₁ in the model provided.

Response: We thank the reviewer for this comment. We have checked map density and found that S218^{ICL2} indeed points away from G₁₁ in the model and should not interact with the G₁₁ protein. We have deleted this comment on “S218^{ICL2} formed hydrophobic interactions with G₁₁” in the main text.

Minor point. The number of significant figures shown for data in tables is likely too high in many cases.

Response: We appreciated this comment and the number of significant figures shown for data in tables has been revised.

REVIEWER COMMENTS

Reviewer #1 (Remarks to the Author):

The authors have answered all the questions I raised satisfactorily and I can now recommend publication of the manuscript in Nat Communications. However, I still have some minor things that should be fixed before publishing:

1. Lines 117-118: There is an additional structure known for a GPCR-G11 complex: PDB 7RKF: US28-G11 (Tsutsumi et al., 2022). This one should be listed as well.
2. Fig. 2A: the names of the G proteins overlapp with the ECD and are hard to read.
3. The English should be checked again. There are some grammar mistakes: e.g. Line 142: "The $\alpha 5$ helix of Go are rotated $\sim 8.2^\circ$ ", where singular should be used.

Reviewer #3 (Remarks to the Author):

The authors have addressed most of my concerns; however, the map quality does not justify many of the side chains and some of the backbone in the G11 complex. I agree that the use of Rosetta should improve quality of the models, but is shouldn't be used to add side chains where there is no map density and then deposit that model. The average reader who downloads the models will assume that they are based on the maps. If you want to use Rosetta to predict side chains for which there is no density for the purpose of discussion, such as the role of K258, you should acknowledge in the text that this side chain is based on Rosetta refinement, not the map. In this region of the receptor (K258-L263) the map doesn't even justify modeling some of the backbone. I believe the readers of Nature Communications will expect better from this well-established structural biology group. Therefore, I cannot recommend acceptance until the models are modified to reflect the quality of the maps.

Point-by-Point Responses to the Comments Made by the Reviewers

Reviewer #1 (Remarks to the Author):

The authors have answered all the questions I raised satisfactorily and I can now recommend publication of the manuscript in Nat Communications. However, I still have some minor things that should be fixed before publishing:

1. Lines 117-118: There is an additional structure known for a GPCR-G₁₁ complex: PDB 7RKF: US28-G₁₁ (Tsutsumi et al., 2022). This one should be listed as well.

Response: We thank the reviewer for this valuable suggestion. We have modified the text to add an additional structure known for a GPCR-G₁₁ complex and cited the paper of Tsutsumi et al., 2022. (References 36) in the revised paper.

2. Fig. 2A: the names of the G proteins overlapped with the ECD and are hard to read.

Response: We thank the reviewer for the careful reading. We have made a modification in the Fig. 2A of the revised manuscript.

3. The English should be checked again. There are some grammar mistakes: e.g. Line 142: "The $\alpha 5$ helix of G_o are rotated $\sim 8.2^\circ$ ", where singular should be used.

Response: We appreciate the reviewer for the careful reading and constructive suggestions. We have corrected the grammar mistakes in the revised manuscript.

Reviewer #3 (Remarks to the Author):

The authors have addressed most of my concerns; however, the map quality does not justify many of the side chains and some of the backbone in the G₁₁ complex. I agree that the use of Rosetta should improve quality of the models, but it shouldn't be used to add side chains where there is no map density and then deposit that model. The average reader who downloads the models will assume that they are based on the maps. If you want to use Rosetta to predict side chains for which there is no density for the purpose of discussion, such as the role of K258, you should acknowledge in the text that this side chain is based on Rosetta refinement, not the map. In this region of the receptor (K258-L263) the map doesn't even justify modeling some of the backbone. I believe the readers of Nature Communications will expect better from this well-established structural biology group. Therefore, I cannot recommend acceptance until the models are modified to reflect the quality of the maps.

Response: We thank the reviewer for this comment. We agree that we should not include the side chains or the loop regions with no density in the final models. We therefore further examined the model and truncated more than 80 side chains (including those truncated in the first revision) and the loop regions (D103-K105, and G261-D262) in the UCN1-CRF2R-G₁₁ structure. We believe that side chains refined by Rosetta represent the chemically optimized rotamers, and would like to keep those key interface residues in our discussion, which we indicated in the relevant figure legends, and added a short paragraph in the main text. Please see the paragraph in the section of "Cryo-EM structure determination of UCN1-CRF2R-G₁₁ and UCN1-CRF2R-G_o complexes": Both structures were carefully examined and refined with real space and Rosetta refinement techniques against the cryo-EM maps. The final model of the CRF2R-G₁₁ complex has many side chains truncated due to the lack of density. We used side chain rotamers from the Rosetta-refined model for the purpose of discussion, which are indicated in the relevant figure legends. The revised structure coordinates has been deposited to the PDB database.

REVIEWER COMMENTS

Reviewer #3 (Remarks to the Author):

I'm disappointed to see that the authors did not significantly improve their model. Their 'revised model' still has numerous side chains that are modeled in the absence of adequate map density in both receptor and G protein. It is also not clear how they modeled the fatty acids given the map quality (see below). I don't understand the authors reluctance to provide a model that reflects the quality of the maps. This reflects poorly on them and on Nature Communications if the manuscript is accepted as is. As I indicated in my previous review, I was willing to accept the manuscript if they submitted a model that adequately reflected the quality of the maps. As is stands, I cannot recommend acceptance and I would prefer not to review the manuscript again.

Point-by-Point Responses to the Comments Made by the Reviewers

Reviewer #3 (Remarks to the Author):

I'm disappointed to see that the authors did not significantly improve their model. Their 'revised model still has numerous side chains that are modeled in the absence of adequate map density in both receptor and G protein. It is also not clear how they modeled the fatty acids given the map quality (see below). I don't understand the author's reluctance to provide a model that reflects the quality of the maps. This reflects poorly on them and on Nature Communications if the manuscript is accepted as is. As I indicated in my previous review, I was willing to accept the manuscript if they submitted a model that adequately reflected the quality of the maps. As it stands, I cannot recommend acceptance and I would prefer not to review the manuscript again.

Response: We thank the reviewer for his/her comments. In this revision, we have further deleted the lipid molecules and further truncated the side chains which have no adequate electron density. A total of 240 side chains in the receptor and the urocortin ligand, about 100 side chains in the trimeric G protein, and 24 residue chains in the nanobody, have been truncated. The PDB entry (7TRY) has been updated with this revised model. While we updated the PDB deposit with this truncated model to reflect the map quality, we have kept the Rosetta refined side chains in our discussion in the manuscript, which is indicated in the section of "Cryo-EM structure determination" in main text, in the methods section, in the relevant figure legends, and in the Data Availability statement of the manuscript. Rosetta refinement has been used to refine cryo-EM structure with near-atomic resolution, and has been proven to optimize global protein geometry and side chain rotamers, thus greatly improve model accuracy of cryo-EM structures (R. Y.-R. Wang, et al. 2016 <https://doi.org/10.7554/eLife.17219>). Rosetta refined models are, therefore, acceptable in many structural papers for discussion, with structure resolutions far worse than the 3.7 Å resolution reported in our paper here. Based on the PDB validation report, the quality of our structures is in top 10 percentile. We hope that our exhaustive efforts have fulfill the reviewer's stringent criteria on the quality of the structures.

REVIEWERS' COMMENTS

Reviewer #4 (Remarks to the Author):

This manuscript reports single particle cryo-EM structures of 1) CRF2R bound to UCN1 in complex with Go and 2) CRF2R bound to UCN1 in complex with G11. Several of the figures and conclusions compared models built into these cryo-EM maps with the previous determined cryo-EM structure of CRF2R coupled to Gs (PDB: 6PB1) and M1R-G1 (PDB: 6O1J) and discuss how the differences in the models built into these cryo-EM structures provide exciting insights into how CRF2R is able to couple with multiple G protein subtypes.

Essential Revisions:

The authors built models into these cryo-EM maps with a starting model from their previously determined structure of CRF2R in complex with Gs (PDB: 6PB1). For UCN1-CRF2R-G11, the manually adjusted model was then refined using Phenix and automated Rosetta refinement. Automated Rosetta refinement (Wang et al 2016 eLife) helped refine the backbone and side chain geometries of the manually rebuilt models into the cryo-EM maps. However, in the regions of the model where side chain density is too weak to unambiguously assign a conformation it is essential that the residues be stubbed to the C β position in both the truncated and Rosetta-refined model deposited as shown in the μ OR-Gi complex (PDB: 6DDE; Koehl et al., 2018 Nature).

In the both the truncated and Rosetta-refined models of UCN1-CRF2R-G11 there remain some regions where it is not possible to assign a conformation for the residues (e.g. ECD). The residues should be stubbed to the C β position.

While there is clearly structured density outside the TMs of CRF2R in the UCN1-CRF2R-Go cryo-EM map, the CLRs and PLMs modeled into this density is not mentioned in the text and should be deleted from the UCN1-CRF2R-Go model.

For the figures using the Rosetta Model with non-stubbed residues (e.g. Fig 4c) it would be useful to overlay this over the density as a surface mesh to aid interpretation.

Minor points:

1) Line 70: UNC should read UCN

2) Figure 2b and d: ELC should read ECL

3) Figure 3c: G11/q should read G q/11

4) Figure 5: The color-coded figure legend in Fig 2a and 4a is useful for comparing the models. This would also be helpful in Figure 5 when comparing CRF2R-G11 with M1R-G11.

5) Line 158: Should read "The third and fourth to last"

6) Line 166: "c" should read couple

7) Line 372: should read "were truncated"

8) Line 372: should read "panel were from"

REVIEWERS' COMMENTS

Reviewer #4 (Remarks to the Author):

This manuscript reports single particle cryo-EM structures of 1) CRF2R bound to UCN1 in complex with Go and 2) CRF2R bound to UCN1 in complex with G11. Several of the figures and conclusions compared models built into these cryo-EM maps with the previous determined cryo-EM structure of CRF2R coupled to Gs (PDB: 6PB1) and M1R-G1 (PDB: 6O1J) and discuss how the differences in the models built into these cryo-EM structures provide exciting insights into how CRF2R is able to couple with multiple G protein subtypes.

Response: We thank the reviewer for his/her positive assessment on the importance and quality of our paper. We also appreciate for his/her very constructive suggestions. Responses to his/her specific comments are detailed below.

Essential Revisions:

The authors built models into these cryo-EM maps with a starting model from their previously determined structure of CRF2R in complex with Gs (PDB: 6PB1). For UCN1-CRF2R-G11, the manually adjusted model was then refined using Phenix and automated Rosetta refinement. Automated Rosetta refinement (Wang et al 2016 eLife) helped refine the backbone and side chain geometries of the manually rebuilt models into the cryo-EM maps. However, in the regions of the model where side chain density is too weak to unambiguously assign a conformation it is essential that the residues be stubbed to the C β position in both the truncated and Rosetta-refined model deposited as shown in the μ OR-Gi complex (PDB: 6DDE; Koehl et al., 2018 Nature).

In the both the truncated and Rosetta-refined models of UCN1-CRF2R-G11 there remain some regions where it is not possible to assign a conformation for the residues (e.g. ECD). The residues should be stubbed to the C β position.

Response: We thank the reviewer for this comment. Based on the suggestions from the previous reviewer and the recommendation of the editor, we will submit two version of PDB: one version is with truncated version of PDB, which side chains with weak density will be stubbed to the C β position, and this version will be deposited to the PDB database; the other version is the Rosetta-refined model with side chains built into the model to illustrate the potential interaction of the receptor-G protein interface in the CRF2R-UCN1-G₁₁ complex.

While there is clearly structured density outside the TMs of CRF2R in the UCN1-CRF2R-Go cryo-EM map, the CLRs and PLMs modeled into this density is not mentioned in the text and should be deleted from the UCN1-CRF2R-Go model.

Response: We thank the reviewer for this comment. Based on the reviewer's comments, we have again inspected the truncated model of CRF2R-UCN1-G₁₁, and further stubbed the side chains of 13 receptor residues (to C β) with weak density, among which six residues are from the ECD. We have updated the PDB entry of this structure (7TRY) with its PDB report.

For the figures using the Rosetta Model with non-stubbed residues (e.g. Fig 4c) it would be useful to overlay this over the density as a surface mesh to aid interpretation.

Response: We thank the reviewer for this valuable suggestion. We have added additional supplementary

figures in Supplementary Fig. 4c-f, which are overlaid the density as a surface mesh to aid interpretation of all figures using the Rosetta Model with non-stubbed residues (e.g. Fig 2c, 2g, Fig 3h, Fig 4d, Fig 5b, 5d,5 f).

Supplementary Fig. 4 Cryo-EM density maps of the UCN1-CRF2R–G protein structures.

(c) The density are shown as a surface mesh for K258^{ECL2} (cornflower blue), which was modeled with Rosetta in the UCN1-CRF2R-G₁₁ structure and forms an H-bond with P3^{UCN1} (coral). (d) The density are shown as a surface mesh for the side chains of K372, and D379 of the receptor (cornflower blue), and R42 and D312 of Gβ1 (aquamarine) in the UCN1-CRF2R-G₁₁ complex, which were truncated in the structure and whose rotamers shown in this panel were from the Rosetta-refined model. (e) Receptor side chains and those of K345, D346, and E355 on Gα-α5 (hot pink) were truncated in the UCN1-CRF2R-G₁₁ structure, whose rotamers shown in this panel were from the Rosetta-refined model and their density are shown as a surface mesh. (f) The side chains of receptor residues, and N198, I199, and K345 of G₁₁ in this panel were truncated in the UCN1-CRF2R-G₁₁ structure. Those residues shown here were prepared based on the Rosetta-refined model and the density are shown as a surface mesh to aid interpretation of the interactions between ICL2 and G₁₁ (hot pink).

Minor points:

- 1) Line 70: UNC should read UCN
- 2) Figure 2b and d: ELC should read ECL
- 3) Figure 3c: G11/q should read G q/11
- 4) Figure 5: The color-coded figure legend in Fig 2a and 4a is useful for comparing the models. This would also be helpful in Figure 5 when comparing CRF2R-G11 with M1R-G11.
- 5) Line 158: Should read “The third and fourth to last”
- 6) Line 166: “c” should read couple
- 7) Line 372: should read “were truncated”
- 8) Line 372: should read “panel were from”

Response: We are very grateful for careful reading and detailed review by the reviewer. We have corrected these typographical and grammatical errors in the revised manuscript.